# NeRT: Implicit Neural Representation for Time Series

## Abstract

Time series is one of the most fundamental data types in real-world environments and there have been many different deep-learning models to effectively handle time series data, ranging from recurrent neural networks to Transformers to differential-equation-based models. These existing models, however, tend to underperform due to irregular measurements, sensitivity to hyper-parameters (e.g., a window size), to name a few. Modeling time series as a continuous-in-time signal via implicit neural representations (INRs) can be an alternative approach to overcome such limitations. However, naïve adoptions of existing INR frameworks toward time series do not yield promising outcomes. To address this, we propose NeRT, a novel class of INRs tailored to handle time series data; the core ideas are to design a new coordinate system, to employ learnable Fourier features, and to model periodic and scale components of time series separately. Thanks to the inherent characteristics of INRs, our model can learn from both regular and irregular time series in a continuous-time manner and perform time series forecasting and imputation at the same time with a single trained model. Moreover, we show that NeRT can be efficiently parameterized via latent modulation. Through extensive experiments with real-world and scientific datasets, we demonstrate that NeRT significantly outperforms baselines including popular INR-based methods and previous time series models.

## 1 Introduction

Time series processing is one fundamental task of machine learning. Since time series can be observed frequently in our daily life, ranging from stock prices to weather conditions, it is of utmost importance to process time series data appropriately. Among many tasks related to it, time series forecasting and imputation are two rudimentary tasks in the field of time series processing: i) many real-world applications are basically time series forecasting, e.g., weather forecasting, and ii) collected time series data often accompany missing observations and we need to impute them before processing.

To this end, many different deep-learning (DL) algorithms have been proposed so far. Recurrent neural networks (RNNs) and their variants (e.g., long short-term memory, or LSTM) (Connor et al., 1994; Hochreiter & Schmidhuber, 1997; Qin et al., 2017; Lai et al., 2018; Sherstinsky, 2020) have been (one of) the first DL algorithms for processing time-series data, but are typically limited to handle regularly-sampled time series. Transformers (Vaswani et al., 2017; Zhou et al., 2021; Wu et al., 2021; Liu et al., 2021; Wen et al., 2022; Zhou et al., 2022) quickly superseded RNNs thanks to its higher representation learning capability aided by the self-attention. They, however, typically share the same limitation with RNNs, reliance on regularly-sampled time-series data, and also require a large dataset as the model tend to consist of millions of model parameters.

Alternative approaches that are capable of handling irregularly-sampled time series include the differential equation-based DL paradigm, also known as *continuous-time* methods. Neural ordinary differential equations (NODEs) (Chen et al., 2018) and neural controlled differential equations (NCDEs) (Kidger et al., 2020) are two exemplary works in this line of research. These models can be lightweight in terms of memory usage, but are likely to be heavyweight in terms of computational costs and the training/inference time. This is because they require numerically solving the initial value problems with a time integrator; expensive higher-order ODE solvers, such as the Dormand–Prince solver (Dormand & Prince, 1980), are typically used for the sake of accuracy.

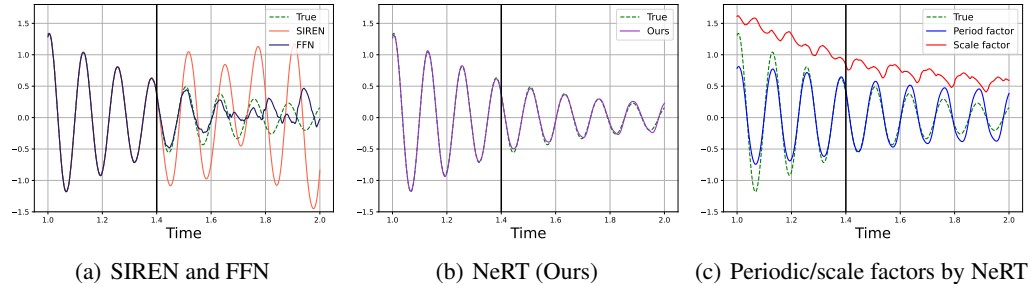

(a) SIREN and FFN     (b) NeRT (Ours)     (c) Periodic/scale factors by NeRT

Figure 1: **Preliminary study with a damping oscillatory signal.** Extrapolation results (Figures 1 (a)-(b)) and extracted factors during training (Figure 1 (c)). The left side of the solid vertical line represents the training range, while the right side represents the testing range.

In this paper, to overcome the limitations of the existing time-series methods we propose a method called implicit **Ne**ural **R**epresentation for **T**ime series (NeRT) based on the implicit neural representation (INR) paradigm. The proposed method does not require regularly-sampled data, millions of model parameters, and numerical computation for training/inference. Moreover, NeRT is free from the concept of a sliding window (which is common in nearly all existing methods) and, thus, requires only a single training for performing both forecasting and imputation. To accomplish the goal, NeRT i) operates on a spatiotemporal coordinate system of multi-variate time-series data, ii) learns a Fourier feature mapping for the coordinate system, and iii) generates periodic and scale components of time series. To our knowledge, this is the first work to identify limitations of existing DL models in time series domain and systematically adopt INRs to time series modeling.

## 2 WHY INRS? – LIMITATIONS OF EXISTING DL METHODS

In this section, we first discuss the limitations of existing time series modeling approaches, which motivates us to develop a novel INR-based time-series modeling approach. Here, we identify four major limitations: **L1**) difficulties in handling irregular time series, **L2**) heavy reliance on the input/output window size, **L3**) non-existence of unified model for time series forecasting and imputation, and **L4**) model scalability, computation/memory-intensive, and insufficient training data.

Firstly, processing irregular time series is one of the most challenging problems to address in the field of time series processing. The irregularity of time series arises mainly due to i) missing values, ii) misaligned cycles of variables in multi-variate time-series, iii) event-driven sensing, and so forth. Some remedies include time series embedding (Kazemi et al., 2019), positional encoding (Vaswani et al., 2017), padding, or likelihood-based approaches (Mei & Eisner, 2017), which partially resolves the difficulty at increased cost. The alternative methods, continuous-time models, naturally provides a formalism to handle irregularly-sampled data (Chen et al., 2018; Kidger et al., 2020), but achieving high-performance is often challenging (due to the second limitation below).

Secondly, nearly all time-series DL models operate on a sliding window (i.e., reading $m$ past observations and predicting $n$ observations ahead, where $m$ and $n$ are hyperparameters). The performance of the models is highly sensitive to those hyperparameters (cf. Appendix B) and also changing the window sizes requires retraining a model from the scratch.

Thirdly, relevant to the second point, most time series models can only perform a single task. In other words, for imputation and forecasting, two different models should be trained from scratch. This is indeed very inefficient considering that time series imputation (interpolation) and forecasting (extrapolation) typically share many common characteristics.

Lastly, it requires a sufficient amount of data for model training (Shorten & Khoshgoftaar, 2019; Wen et al., 2020) to extract features/patterns correctly. In real-world scenarios, however, data scarce scenarios are common, which makes training a time series model challenging; large models are quickly overfitted and their testing accuracy become mediocre. For instance, Transformer-based time series models are sometimes surprisingly worse than simple Linear models (Zeng et al., 2022). Also,

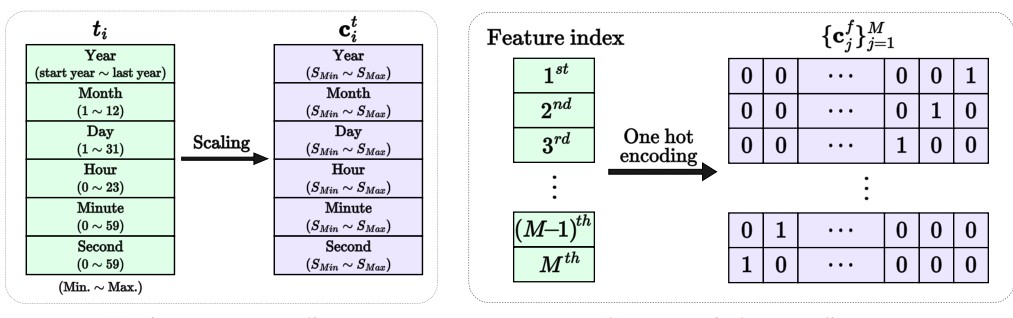

(a) Time stamp encoding             (b) Feature index encoding

Figure 2: **Spatiotemporal coordinate construction.** Our method to define temporal coordinates $\{\mathbf{c}_i^t\}_{i=1}^N$ (Figure 2 (a)) and spatial coordinates $\{\mathbf{c}_j^f\}_{j=1}^M$ (Figure 2 (b)).

taking a forward pass of a model could be computation extensive (due to many model parameters in Transformers and complex numerical integration in continuous-time models).

**INRs:** INRs learn a continuous representation of a signal, which are collected in discrete measurements, not necessarily sampled in a uniform mesh grid. Being independent on a regularly-sampled data and learning the continuous signal naturally address difficulties in model training on irregularly sampled data (not even on a Cartesian system, addressing **L1**). Also, as INRs learn a mapping from a coordinate to a signal at that coordinate, training and inference can be performed by providing a set of coordinates, which enables a sliding-window-less imputation and forecasting (addressing **L2** and **L3** simultaneously). Finally, INRs operate on a per-data-instance basis, meaning that one time-series instance is required to train an INR (addressing **L4**).

Simply reusing existing INRs, however, results in poor performance even for learning a simple and noise-less simulated time series data (See Figure 1 for results and Appendix D for setups). We test two well-known INRs, sinusoidal representation networks (SIRENs) (Sitzmann et al., 2020) and Fourier feature networks (FFNs) (Tancik et al., 2020), for performing the time series forecasting task, testing coordinates are selected outside a training region. Both SIREN and FFN are known to learn very complex (high-frequency) signals (e.g., images with details) by using either the sinusoidal activation or Fourier features. However, extrapolation capability of such INRs have not been investigated.

## 3 DESIDERATA FOR INR-BASED MODELING FOR TIME SERIES

Now, we identify some desired characteristics required for INRs in handling time-series data and based on the identified characteristics we design an efficient and effective INR-based method, called NeRT, for time series forecasting and imputations.

### 3.1 SPATIO-TEMPORAL COORDINATE SYSTEMS OF TIME SERIES DATA

Unlike domains of applications where the coordinate system is relatively well-defined (i.e., 2d Cartesian coordinate systems for images or solutions of partial differential equations, PDEs), multi-variate time-series modeling needs a new definition of a coordinate system. Given an $M$-dimensional multi-variate time series $\{\mathbf{x}_i\}_{i=1}^N$, where $\mathbf{x}_i = \mathbf{x}(t_i) \in \Omega^M$, a naïve way of building INRs is to directly use the time stamps, $t_i$, as a coordinate and $\mathbf{x}_i$ as a signal intensity. However, such a coordinate system is too primitive to properly represent multi-variate time-series data in INRs.

Instead, we propose to manufacture a refined coordinate system, providing fine-granularity in indexing temporal domain and feature domain (See Figure 2). We interpret the feature domain, i.e., the $M$-dimensional space, $\Omega^M$, as a spatial domain and, thus, propose a novel spatio-temporal coordinate system for multi-variate time-series data. In our new proposed coordinate system, a coordinate can be expressed as $\{\mathbf{c}_i := \mathbf{c}_i^t, (\{\mathbf{c}_j^f\}_{j=1}^M)\}_{i=1}^N$, where $\mathbf{c}_i^t \in \mathbb{R}^{D_{\mathbf{c}^t}}$ and $\mathbf{c}_j^f \in \mathbb{R}^{D_{\mathbf{c}^f}}$. Here, $\mathbf{c}_i^t$ and $\mathbf{c}_j^f$ denote the temporal and the spatial coordinates of $t_i$ ($i \in N$) and the $j$-th feature ($j \in M$), respectively.

For the temporal coordinate, we apply a simple min-max scaling to put different quantitative values (year, month, day, etc) into the same numerical scale $[S_{\min}, S_{\max}]$, effectively resolving numerical issues. For the spatial coordinate, we employ one-hot encoding, which translate an integer index into one-hot vectors. Unlike images or PDE solutions, the spatial locality in the feature domain is less clear. Moreover, imposing a certain order based on the integer index is likely to impose undesirable biases. Thus, we propose to use one-hot encoding, which removes reliance on a specific ordering and is general enough to represent represent many different multi-variate time series.

## 3.2 FOURIER FEATURES OF INRS

Fourier features of signals introduce a new angle, interpreting the signals in the frequency domain, which is known to be effective in time-series modeling. Thus, using Fourier features in the design of time-series INRs can be evidently beneficial for time-series modeling. Recently proposed INRs, such as SIRENs and FFNs, also aggressively use Fourier features for modeling signals mainly from computer vision tasks. Both methods commonly discuss the effectiveness of Fourier features; based on the neural tangent kernel (NTK) theory (Jacot et al., 2018), they showed that ordinary fully-connected (FC) networks have spectral biases, i.e., lower-frequency information is learned earlier than higher-frequency ones, and that using Fourier features alleviates this spectral bias problem.

However, those existing Fourier feature extraction methods are optimized mainly toward images (e.g., a hand-tuned frequency in the input layer of SIREN) and they do not show satisfactory performance for time series (See again Figure 1 and comparisons in the experimental section). We overcome the limitation by learning a Fourier feature mapping layer (like FFNs but we learn a Fourier mapping instead of the FFNs' hand-craft mapping) and utilizing the sinusoidal activation (like SIRENs) — the details of our model design for NeRT will be described shortly in the next section and our theoretical analyses justify the appropriateness of our design to time series processing.

## 3.3 DECOMPOSITION OF TIME SERIES INTO INTERPRETABLE FACTORS

Decomposing a time series signal into interpretable factors, e.g., seasonality and trend, is a common practice for improving the performance of models in the field of time series (Cleveland et al., 1990) (cf. Section 6). Separately predicting them and later combining them into a signal is effective in improving the time series model accuracy (Hamilton, 2020). We propose a new INR design for separately modeling the periodic factor and the scale factor and later combining them to reconstruct time-series signal. This factor-by-factor processing approach alleviates the modeling burden of INRs for time series. Also, we do not explicitly supervise the learning of the two individual factors; instead we train NeRT with the original undecomposed time series signal to reduce the training overhead and implicitly learn the two factors. Our theoretical analysis (cf. Appendix C) shows that learning time series through NeRT allows effectively extracting periodic factors.

# 4 MODEL ARCHITECTURE

We describe our proposed INR-based framework (cf. Figure 3), NeRT, for time series forecasting and imputation. As we intend to learn a time-series signal in a factored form, a periodic factor and a scale factor, we design an encoder-decoder-type neural network architecture, where (i) the encoders generate embeddings for input spatiotemporal coordinates, and ii) the decoders take the embeddings as input and produce a periodic factor and a scale factor, individually. The two factors are then multiplied to produce the signal at the input coordinate.

## 4.1 ENCODER

The encoder of NeRT reads the spatiotemporal coordinate defined in Section 3.1; given $\{(\mathbf{x}_i, t_i)\}_{i=1}^N$, we apply scaling and one-hot encoding to obtain the spatiotemporal coordinates $\{\mathbf{c}_i := \mathbf{c}_i^t, (\{\mathbf{c}_j^f\}_{j=1}^M)\}_{i=1}^N$ and feed these coordinates to the encoder.

**Embedding of the spatiotemporal coordinate** We then use the following FC layers with sinusoidal activations, i.e., SIREN, for a spatiotemporal coordinate $(\mathbf{c}_i^t, \mathbf{c}_j^f)$ — we note that this coordinate

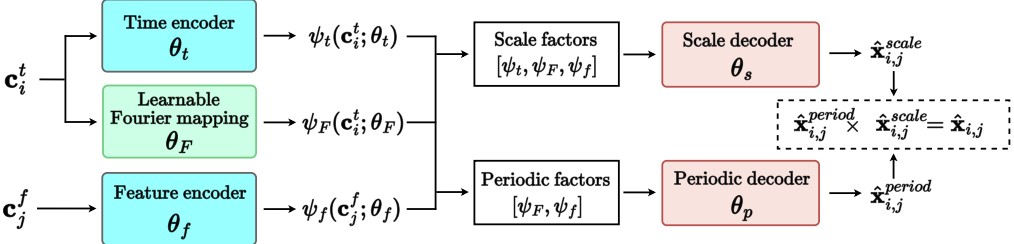

Figure 3: **NeRT architecture.** The spatiotemporal coordinate $(\mathbf{c}_i^t, \mathbf{c}_j^f)$ is converted to periodic/scale factors through $\psi_t, \psi_F$, and $\psi_f$, followed by two decoders $\phi_s$ and $\phi_p$ to effectively infer the signal intensity $\mathbf{x}_{i,j}$ at the spatiotemporal coordinate.

represents the minimum processing granularity, i.e., $j$-th feature of $\mathbf{x}_i$ at time $t_i$:

$$\psi_t(\mathbf{c}_i^t; \theta_t) := FC_{L_t}(\rho_r(FC_{L_t-1} \cdots (\rho_r(FC_1(\mathbf{c}_i^t))))),$$
$$\psi_f(\mathbf{c}_j^f; \theta_f) := FC_{L_f}(\rho_r(FC_{L_f-1} \cdots (\rho_r(FC_1(\mathbf{c}_j^f))))),$$
(1)

where $L_t$ and $L_f$ mean the number of layers in each module respectively, and $\rho_r$ is ReLU. Since $\mathbf{c}_j^f$ is an one-hot vector, one can consider that $\psi_f(\mathbf{c}_j^f; \theta_f)$ generates an embedding for the one-hot vector.

**Learnable Fourier feature mapping**    The previous two embeddings, $\psi_t$ and $\psi_f$, do not generate Fourier features and therefore, we use the following method to generate them. In particular, our method is to learn an optimal Fourier mapping layer (instead of hand-crafted ones):

$$\psi_F(\mathbf{c}_i^t; \theta_F) := \{\mathbf{A}_m \odot \sin(\boldsymbol{\omega}_m \cdot \mathbf{c}_{i,m}^t + \delta_m) + \mathbf{B}_m\}_{m=1}^{D_{\mathbf{c}^t}}, \quad \boldsymbol{\omega}_m \sim \mathcal{U}(a, b)$$
(2)

where $\mathbf{c}_{i,m}^t$ is $m$-th value of $\mathbf{c}_i^t$ and $\odot$ denotes elementwise multiplication. $\boldsymbol{\omega}_m \in \mathbb{R}^{1 \times D_{\psi_F}}$ is the $m$-th frequency of the Fourier feature $\psi_F$, which is sampled from a uniform distribution $[a, b]$. A learnable vector $\mathbf{A}_m \in \mathbb{R}^{1 \times D_{\psi_F}}$ indicates the amplitude, initialized to 1. Vectors $\mathbf{B}_m \in \mathbb{R}^{1 \times D_{\psi_F}}, \delta_m \in \mathbb{R}^{1 \times D_{\psi_F}}$ denote the phase shift and the bias respectively, and they are initialized to 0. Optionally, we can employ an additional FC layer after Eq. 2 to compress the embedding. All together, we use three types of embeddings, $\psi_t, \psi_f$, and $\psi_F$ given the spatiotemporal coordinate.

**Remark 1** *According to the NTK theory (Jacot et al., 2018), our kernel satisfies the stationary and the shift-invariant properties — therefore, one can consider that the learnable Fourier feature mapping of NeRT performs a 1D convolution-based processing of time series with a learnable kernel. In addition, our proposed mapping has an additional property that resorts to the extreme value theorem to perform time series forecasting, i.e., extrapolation. In other words, $\psi_F$ is i) continuous, and ii) confined to the min/max values defined by $\mathbf{A}$.*

## 4.2   DECODER

The proposed decoder architecture separately generates the periodic and the scale factors, denoted $\phi_p$ and $\phi_s$, and multiply them to infer the signal intensity:

$$\phi_p(\psi_F \oplus \psi_f; \theta_p) := \rho_s(FC_{L_p}(\rho_s(FC_{L_p-1} \cdots (\rho_s(FC_1(\psi_F \oplus \psi_f)))))),$$
$$\phi_s(\psi_F \oplus \psi_f \oplus \psi_t; \theta_s) := FC_{L_s}(\rho_r(FC_{L_s-1} \cdots (\rho_r(FC_1(\psi_F \oplus \psi_f \oplus \psi_t))))),$$
(3)

where $L_p$ and $L_s$ are the numbers of layers in the two decoders, $\rho_s$ is the sinusoidal function, and $\oplus$ denotes a concatenation of vectors. The output of $\phi_p$ and $\phi_s$ correspond to $\hat{\mathbf{x}}^{period}$ and $\hat{\mathbf{x}}^{scale}$ in Figure 3. We constrain $\hat{\mathbf{x}}_{i,j}^{period} \in [-1, 1]$ by using the Sine activation and $\hat{\mathbf{x}}_{i,j}^{scale} \in \mathbb{R}$ to denote the inferred periodic and scale factors, respectively. Note that the decoder to infer the periodic factor, i.e., $\phi_p$, mainly rely on our Fourier feature $\psi_F$; since $\psi_f$ is the embedding of the spatial coordinate, only $\psi_F$ contains the temporal information to infer. The scale decoder reads all available embeddings for the input coordinate $(\mathbf{c}_i^t, \mathbf{c}_j^f)$. Our final inference outcome, i.e., the signal intensity, at the spatiotemporal coordinate is $\hat{\mathbf{x}}_{i,j} = \hat{\mathbf{x}}_{i,j}^{period} \times \hat{\mathbf{x}}_{i,j}^{scale}$.

The advantages of our design are twofold: i) Our learnable Fourier feature mapping layer is specialized to extract periodic features according to our NTK-based analysis and, thus, it is a sensible design to model the periodic factor of the inferred signal intensity based on the Fourier feature and ii) the periodic and scale factors give us *interpretable* predictions for time series forecasting and imputation.

### 4.3 TRAINING ALGORITHM

To train, we employ a regular gradient-based optimizer to minimize a mean-squared error (MSE) loss over data and predictions such that $\frac{1}{MN} \sum_{i=1}^{N} \sum_{j=1}^{M} (\mathbf{x}_{i,j} - \hat{\mathbf{x}}_{i,j})^2$. Here, we consider $M$-variate time-series measured at $N$ temporal collocation points (See Appendix E for a pseudo-code like algorithm). We emphasize again that the proposed model requires a single model training and use the same trained model to perform both forecasting and imputation while other existing methods require training of different models for each task.

For training INRs for multiple time-series instances, we can choose from two different options: a *vanilla* mode or a *modulated* mode. For the modulation, we adopt an idea of *latent modulation* to our NeRT in Appendix J. To summarize, in the vanilla mode INRs are trained individually and in the modulated mode the shared part is trained via a meta-learning algorithm and the instance-specific part is trained individually.

## 5 EXPERIMENTS

In this section, we evaluate the performance of NeRT on well-known real-world time series datasets, which can be further categorized into periodic time series (Fan et al., 2022), and long-term time series (Zeng et al., 2022). Additionally, to show that the method can be generally applicable to other domains, we test it to solve partial differential equation (PDE) problems, i.e., 2D-Helmholtz equations (McClenny & Braga-Neto, 2020), which exhibit periodic behaviors over a spatial domain.

We use MSE for the evaluation metric and conduct the experiments with three different random seeds and present their mean and standard deviation of evaluation metrics. More detailed descriptions of experiments and additional analyses such as ablation studies are listed in Appendix.

**Experimental environments** We implement NeRT and baselines with PYTHON 3.9.7 and PY-TORCH 1.13.0 (Paszke et al., 2019) that supports CUDA 11.6. The experiments are conducted on systems equipped with Intel Core-i9 CPUs and NVIDIA RTX A6000, A5000 GPUs.

**Baselines for comparison** As baseline models, we consider SIREN (Sitzmann et al., 2020) and FFN (Tancik et al., 2020), the two representative models in the field of INR. For fair comparison, we set the model sizes to be the same. All models are trained with Adam (Kingma & Ba, 2014) with a learning rate of 0.001. In addition, we use eight existing time-series models including Transformer-based and NODE-based models as non-INR baselines (cf. Appendix I). See the full description of the experimental setup in Appendix F.

### 5.1 FORECASTING AND IMPUTATION ON REAL-WORLD TIME-SERIES DATA

Now, we compare the performance of NeRT and baselines on real-world periodic and long-term time series. Experimental results show that the proposed method outperform in both scenarios, even for the long-term time series, which typically has much weaker periodicity than the periodic time series.

### 5.1.1 PERIODIC TIME SERIES

**Experimental setups** For periodic time series experiments, we select four uni-variate time series datasets, i.e., Electricity, Traffic, Caiso, and NP, which are all famous benchmark datasets used in (Fan et al., 2022) and are known to have some periodic patterns. To demonstrate the efficacy on learning the periodic time series, we conduct interpolation and extrapolation tasks on missing blocks, each of which with a length of 500 observations — for hourly observations, 500 observations correspond to three weeks. We consider up to three missing blocks, i.e., 9 weeks. Since INR models are able to solve interpolation and extrapolation simultaneously, both tasks are tested by a single trained

Table 1: **Experimental results on periodic time series.** The experimental results of the same model, presented on the same row, are from a single training, and the best results are reported in boldface.

| Dataset | # blocks | Interpolation | | | Extrapolation | | |
|---|---|---|---|---|---|---|---|
| | | SIREN | FFN | NeRT | SIREN | FFN | NeRT |
| Electricity | 1 | 0.0200±0.0007 | 0.0189±0.0021 | **0.0061±0.0006** | 0.0256±0.0005 | 0.0144±0.0009 | **0.0057±0.0007** |
| | 2 | 0.0182±0.0009 | 0.0144±0.0012 | **0.0057±0.0005** | 0.0233±0.0006 | 0.0142±0.0006 | **0.0077±0.0019** |
| | 3 | 0.0183±0.0004 | 0.0148±0.0013 | **0.0056±0.0006** | 0.0231±0.0001 | 0.0142±0.0005 | **0.0088±0.0008** |
| Traffic | 1 | 0.0174±0.0007 | 0.0140±0.0007 | **0.0057±0.0002** | 0.0176±0.0009 | 0.0113±0.0007 | **0.0050±0.0002** |
| | 2 | 0.0169±0.0003 | 0.0121±0.0001 | **0.0062±0.0000** | 0.0181±0.0004 | 0.0127±0.0003 | **0.0057±0.0003** |
| | 3 | 0.0169±0.0006 | 0.0114±0.0001 | **0.0055±0.0002** | 0.0185±0.0006 | 0.0130±0.0002 | **0.0115±0.0026** |
| Caiso | 1 | 0.0230±0.0008 | 0.0179±0.0013 | **0.0047±0.0012** | 0.0596±0.0018 | 0.0495±0.0045 | **0.0131±0.0014** |
| | 2 | 0.0206±0.0002 | 0.0179±0.0012 | **0.0041±0.0002** | 0.0427±0.0005 | 0.0399±0.0017 | **0.0148±0.0066** |
| | 3 | 0.0337±0.0004 | 0.0326±0.0017 | **0.0099±0.0007** | 0.0260±0.0007 | 0.0196±0.0008 | **0.0051±0.0008** |
| NP | 1 | 0.0510±0.0005 | 0.0502±0.0013 | **0.0149±0.0005** | 0.0326±0.0014 | 0.0346±0.0021 | **0.0235±0.0023** |
| | 2 | 0.0464±0.0004 | 0.0415±0.0002 | **0.0186±0.0008** | 0.0329±0.0010 | 0.0313±0.0015 | **0.0273±0.0022** |
| | 3 | 0.0476±0.0001 | 0.0431±0.0005 | **0.0196±0.0006** | 0.0334±0.0007 | 0.0304±0.0017 | **0.0274±0.0030** |

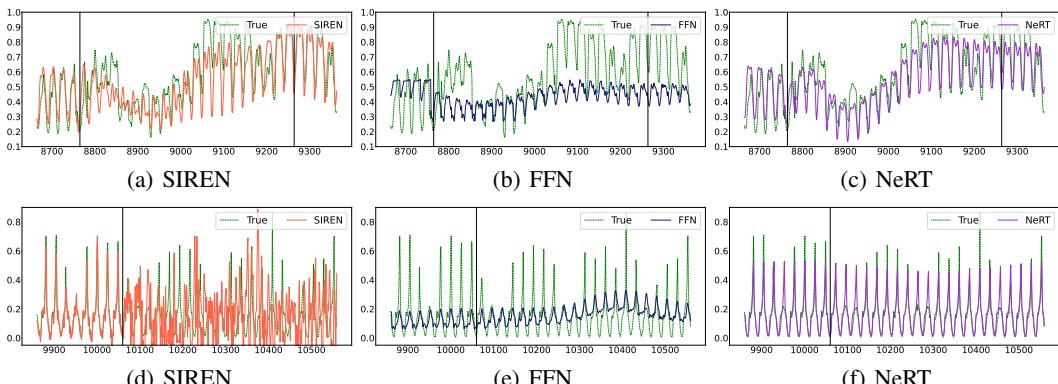

Figure 4: **Forecasting and imputation** [Top] Imputation results in NP ((a)-(c)), the middle area between the two solid vertical lines represents a testing block. [Bottom] Forecasting results in Traffic ((d)-(f)), where the area on the right of the solid vertical line represents a testing range.

model. For example, in the experiment with three blocks, it involves two tasks: i) filling empty non-continuous blocks — in other words, those blocks are scattered in the time domain while each block is contiguous — and ii) forecasting the last 1,500 observations. Detailed experimental settings such as the location of the blocks and hyperparameters are described in Appendix H.

**Comparisons with the existing INRs**   First we present the comparisons against the INR baselines, SIREN and FFN in Table 1, which essentially shows that NeRT outperforms in every task in all scenarios. Especially for Caiso, NeRT exhibits significantly lower errors both in the interpolation and extrapolation tasks, which are around a quarter of those of baselines. Notably, NeRT shows reasonably small MSEs even for extrapolating the last 1,500 observations. Figure 4 visualizes how three models interpolate in NP (Figures 4 (a)-(c)) and extrapolate in Traffic (Figures 4 (d)-(f)). In both cases, only NeRT shows good predictions while two other baselines, FFN and SIREN, struggle. An additional set of experiments with modulated verions of all INRs can be found in Appendix J.

**Ablation studies on the spatiotemporal coordinate**   For all models (SIREN, FFN, and Nert), we test the effect of the proposed coordinate system. In all three models, using the proposed coordinate system is critical in achieving better performance, which is reported in Appendix C.2.1.

**Ablation studies on Fourier feature mapping**   We also study the effect of having learnable Fourier features; the results with learnable or fixed features are reported in Table 4, and demonstrating

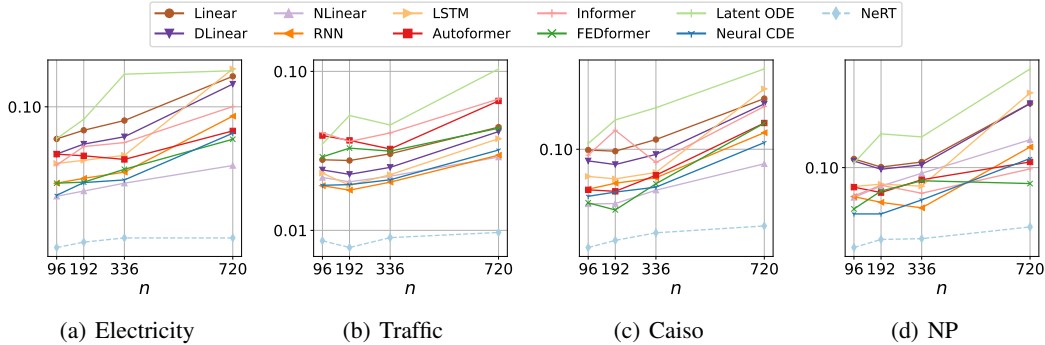

(a) Electricity          (b) Traffic          (c) Caiso          (d) NP

Figure 5: Comparisons with time-series baselines for varying $n = \{96, 192, 336, 720\}$ (output/prediction window size) with a fixed input window size $m = 48$.

through this experiment that adding only a small number of learnable parameters leads to significant performance improvement. We discuss the detailed experimental setup and results in Appendix C.2.2.

**Comparison with non-INR baselines**    To provide a more comprehensive assessment, we conduct experiments comparing NeRT to eight existing time-series models, including popular Transformer-based baselines. (1) Accuracy: In these experiments, we measure forecasting results by varying input/output window size, $m$ and $n$ and the full results are reported in Tables 10, 11 and 12 in Appendix. Figure 5 illustrates the changes in MSE with varying $n$ with $m = 48$. Notably, NeRT consistently achieves lower MSE compared to all other baseline models; particularly, the slope of NeRT MSE is much lower than those of other methods' MSE. We emphasize that NeRT does not require retraining while the other methods need to be retrained for varying $n$. For detailed experimental setups and results, refer to Appendix I. (2) Computational/memory costs: Table 13 in Appendix assures that NeRT has advantages in terms of the overall computational/memory efficiency than the baselines.

### 5.1.2    LONG-TERM TIME SERIES

**Experimental setups**    We conduct experiments on general real-world long-term time series datasets to show the scalability of NeRT. The benchmark datasets of the long-term series forecasting task (Wen et al., 2022) are used for our experiments. Since the periodicity is typically weak in those datasets, we consider this task is much more challenging. We randomly drop 30%, 50%, and 70% observations and evaluate the interpolating performance.

**Experimental results**    Due to space reasons, we list the experiments only with two datasets, ETTh1 and National Illness, in conjunction with their detailed experimental settings. Full results are in Appendix K. According to Table 2, in every drop ratio, NeRT beats other baselines by a large margin. For example, in ETTh1, NeRT shows an MSE of 0.0911, while SIREN and FFN shows 0.2173 and 0.3407, respectively. Moreover, in National Illness, NeRT outperforms other baselines by 489%.

Table 2: Long-term time series

| Dataset | Drop ratio | SIREN | FFN | NeRT |
|---|---|---|---|---|
| ETTh1 | 30% | 0.1945 | 0.2522 | **0.0828** |
| | 50% | 0.2173 | 0.3407 | **0.0911** |
| | 70% | 0.2605 | 0.4256 | **0.1257** |
| National Illness | 30% | 0.3502 | 0.1110 | **0.0239** |
| | 50% | 0.1716 | 0.2319 | **0.0291** |
| | 70% | 0.3564 | 0.4453 | **0.0871** |

### 5.2    SCIENTIFIC PROBLEMS - SOLVING PDES

Here, we demonstrate that NeRT can be applied another domain, namely learning solutions of PDEs — as a matter of fact, PDEs are implicit functions describing dynamics in the spatiotemporal coordinate system. Thus, we test NeRT on learning the solutions of 2D-Helmholtz equation, which is known to produce periodic behaviors. We refer readers to Appendix G for details.

We perform extrapolation tasks over the spatial domain, where the model is trained by using a set of collocation points $(x, y) \in [1, 1.5]^2$ and is tested on $(x, y) \in [1, 2]^2 \backslash [1, 1.5]^2$.

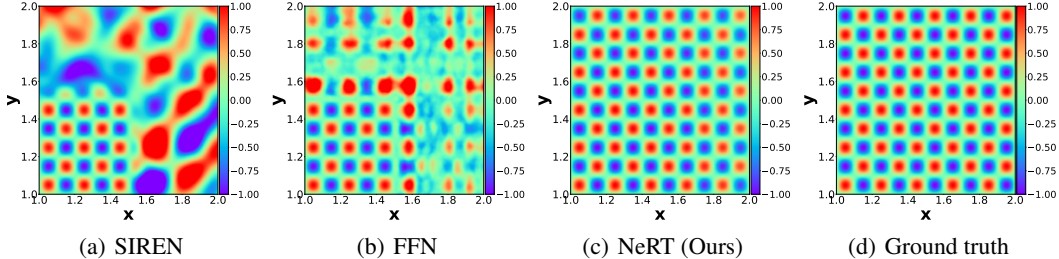

|  (a) SIREN | (b) FFN | (c) NeRT (Ours) | (d) Ground truth |

Figure 6: **Extrapolation task on the 2D-Helmholtz equation.** Results of the extrapolation task with the training range of $x \in [1, 1.5]$ and $y \in [1, 1.5]$, i.e., the left-lower square, (Figures 6 (a)-(c)) and the ground-truth solution (Figure 6 (d)).

**Experimental results**   As shown in Figure 6, all three INR-based models have no difficulties in being overfitted to the training data. However, only the proposed method extrapolates accurately in the test region, while the other two baselines fail. We provide the analyses of the experimental results with other activation functions and additional visualizations in Appendix G.

## 6  RELATED WORK

**Implicit neural representations**   INR approaches, including physics-informed neural networks (PINN) (Raissi et al., 2019) for solving PDE problems and neural radiance fields (NeRF) (Mildenhall et al., 2021) for 3D representation, are quickly permeating various fields. In addition, a counter-measure to the spectral bias (Rahaman et al., 2019) in vanilla multi-layer perceptrons has been proposed, enabling more sophisticated INR-based representations (Sitzmann et al., 2020). According to (Tancik et al., 2020), it is shown that the infinitely wide ReLU based neural network with random Fourier features are equivalent to the shift-invariant kernel method in the perspective of NTK (Jacot et al., 2018). In the time series domain, there exist INR-based studies on unsupervised anomaly detection (Jeong & Shin, 2022) and time series forecasting (Woo et al., 2022). However, none has extensively investigated INRs' applicability for modeling time-series data.

**Decomposition methods used in time series modeling**   Decomposition methods aim to separate a time series sample into multiple components, often including a trend, seasonality, and residual component (Cleveland et al., 1990). These components can then be modeled separately, allowing for more accurate predictions and insights. There exist various decomposition-based methods that have been used for time series, ranging from traditional time series models to recent deep learning models. For example, the wavelet decomposition decomposes time series into different frequency bands (Percival & Walden, 2000; Wang et al., 2018) and the singular spectrum analysis (SSA) decomposes time series into a set of eigenvectors to extract various oscillatory patterns which are interpretable (Vautard & Ghil, 1989; Sulandari et al., 2020).

## 7  CONCLUSIONS

Due to the strength in learning coordinate-based systems, INR has high potential for various fields in natural sciences and engineering. However, despite the promising nature of INR, it has been rarely applied to time series, and no existing unified time series models are based on the INR paradigm. Therefore, we aim to address the limitations that conventional time series models possess and propose NeRT which resorts to the advantages of INR to effectively resolve existing challenges in the field. Based on the INR framework, NeRT effectively learns and predicts time series by separating the periodic and scale factors. Additionally, we suggest a method for embedding the spatiotemporal coordinate of multivariate time series. Through this approach, NeRT clearly outperforms existing INR models and can represent time series more accurately.

## 8 ETHICS STATEMENT

Our model is capable of restoring an incomplete time series sample where some observations are intentionally removed to protect the privacy, e.g., PhysioNet where 90% of observations are removed to hide the identification of patients. Therefore, someone can try data restoration using our method, causing a potential data breach.

## 9 REPRODUCIBILITY STATEMENT

The provided supplementary materials contain the source code and README.md necessary to reproduce our experimental environments and the proposed method. Every source code used in our paper will be made available to contribute to the community.

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

## A    LIMITATIONS

We encode the spatial coordinate of the multivariate time-series as one-hot vectors and transform the temporal coordinate using the min-max scaling. In addition to them, better embedding approaches could be explored as future work.

## B    DRAWBACK OF WINDOW-BASED TIME SERIES MODELS

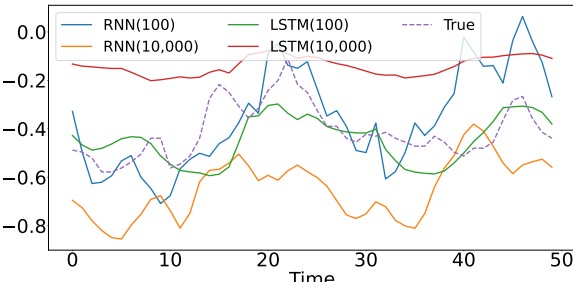

Table 3: Experimental results of time-series models on varying window-sizes

| Model | 100 | 10000 |
|-------|--------|--------|
| RNN | 0.5999 | 0.4478 |
| LSTM | 0.6927 | 1.0530 |

Figure 7: Visualization of results according to window-size of existing time series models

The commonly used approach in time series models, called *shifting window*, assumes a fixed window size. However, this method has several critical drawbacks. First, it requires finding an optimal input window size. Using a too small window size may result in small and inference time, but the model sees short patterns and struggles to infer effectively. On the other hand, employing a too large input window size can enable capturing long-term patterns but at the same time can lead to long training and inference time. Thus, the window size is a highly sensitive hyperparameter, and finding a balance between these two settings is a challenging task.

To empirically show how the size of window affects the model in forecasting time series, we compare results of forecasting 50 future observations by varying the input window size in $\{100, 10000\}$ with other conditions fixed. For the experiment, we choose two typical time series models, RNN and LSTM, and train them for 1,000 epochs on ETTh1. As shown in Table 3, for both RNN and LSTM, their model predictions show big differences depending on the window size, in terms of MSE. Note that models with a small window size, i.e., a window size of 100, shows better predictions when using RNN, while LSTM shows lower MSE when using a longer window size, i.e., a window size of 10000. The same trend can also be observed in Figure 7. In Figure 7, predicted values exhibit significant differences. Therefore, the window size is a critical hyperparameter in modeling time series.

## C THEORETICAL ANALYSES ON INR

### C.1 THEORETICAL ANALYSES

The Fourier feature mapping introduced in (Tancik et al., 2020) transforms the input coordinates using periodic functions, allowing the neural networks to solve the spectral bias in MLPs. This approach is a simple yet powerful way to address the problem. Moreover, the Fourier feature mapping exhibits shift-invariant properties from an NTK perspective. Our learnable Fourier feature mapping enjoys all the advantages of the Fourier feature mapping and moreover, it addresses the difficulty of finding a task-specific fixed set of frequencies in the Fourier feature mapping since we learn them as well. As explained in Section 3, NeRT maps the temporal coordinate onto a desired closed finite interval $[S_{min}, S_{max}]$. Therefore, NeRT can approximate discrete coordinate-based time-series as continuous function in the closed domain and thus, the extrapolation in the original temporal coordinate can be somehow considered as an interpolation in the learned coordinate, e.g., everyday 12pm has $S_{min}$ regardless of year and month. Under these conditions, when coordinates outside the training range are inputted, the domain of NeRT is converted to the maximum and mimimum values of $\psi_F$ by the extreme value theorem. In other words, by Equation 2, NeRT can operate in a wide range of the temporal coordinate, and this approach works very effectively, especially when performing extrapolations.

### C.2 EMPIRICAL ANALYSES

#### C.2.1 ABLATION STUDIES ON THE SPATIOTEMPORAL COORDINATE

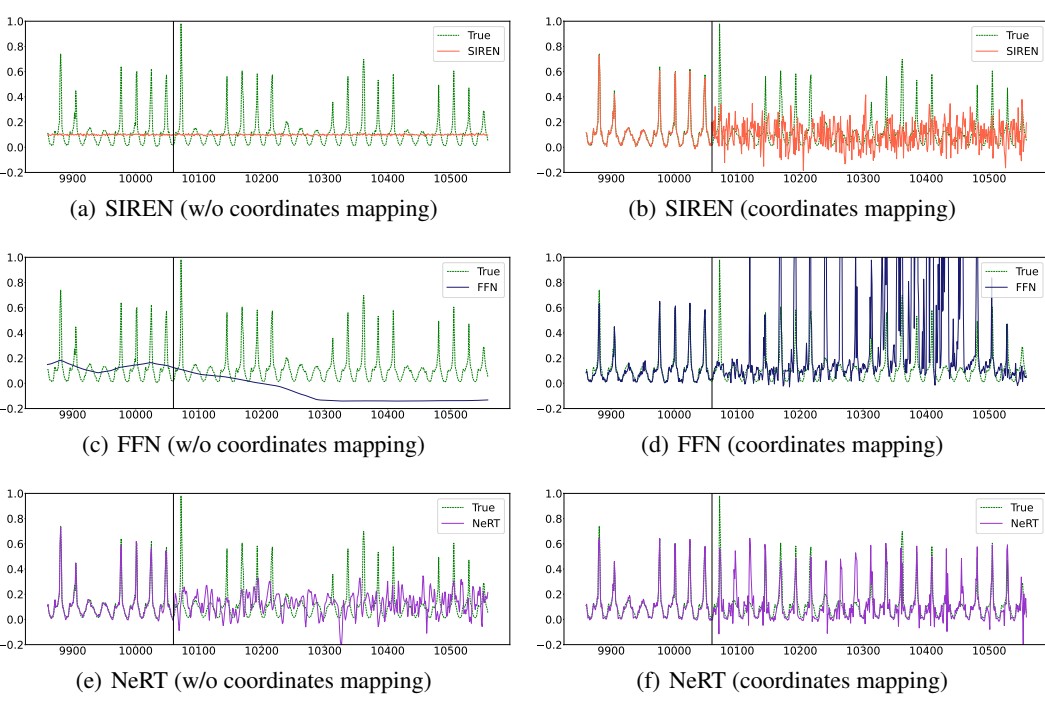

(a) SIREN (w/o coordinates mapping)    (b) SIREN (coordinates mapping)

(c) FFN (w/o coordinates mapping)    (d) FFN (coordinates mapping)

(e) NeRT (w/o coordinates mapping)    (f) NeRT (coordinates mapping)

Figure 8: Experimental results of ablation study according to the spatiotemporal coordinates mapping introduced in Section 3.1. The figures in left column are the results without coordinates mapping and the figures in right column are the results using coordinates mapping.

As an ablation study, we aim to investigate the impact of the proposed coordinates mapping in Figure 2 on INR models. The overall experimental setup follows that of section 5.1. To ensure a fair comparison, we set the model size to be the same across all models. In addition, the initial frequency $w$ of SIREN and NeRT is fixed at 30, which is known to work well in (Sitzmann et al., 2020). The results presented in Figure 8(a), (c), (e) correspond to the models trained without a coordinates

mapping, where a one-dimensional scaled time-stamp used as input to the model. For the baselines, SIREN and FFN, it can be observed that they struggle to accurately represent the overall domain, while NeRT exhibits a more refined representation. However, NeRT still exhibit discrepancies from the ground truth in extrapolation. As shown in Figure 8(b), (d), (f), which is the result of using coordinates mapping, it can be observed that INR models can depict time series more accurately when input coordinates are mapped. In particular, NeRT can even perform sophisticated extrapolation inference.

### C.2.2   ABLATION STUDIES ON LEARNABLE FOURIER FEATURE MAPPING

Table 4: Experimental results of ablation studies on Fourier feature mapping

| Dataset | NeRT (Fixed Fourier mapping) | | NeRT (Learnable Fourier mapping) | |
|---|---|---|---|---|
| | Interpolation | Extrapolation | Interpolation | Extrapolation |
| Electricity | $0.0079\pm0.0015$ | $0.0069\pm0.0009$ | $\mathbf{0.0061}\pm\mathbf{0.0006}$ | $\mathbf{0.0057}\pm\mathbf{0.0007}$ |
| NP | $0.0190\pm0.0016$ | $0.0313\pm0.0044$ | $\mathbf{0.0149}\pm\mathbf{0.0005}$ | $\mathbf{0.0235}\pm\mathbf{0.0023}$ |

To evaluate the effect of learnable Fourier feature mapping, we design experiments comparing its performance with a fixed Fourier feature mapping. The fixed Fourier feature mapping fixes the learnable parameters of the learnable Fourier feature mapping described in Equation 2, specifically, $\omega_m$, $\mathbf{A}_m$, $\mathbf{B}_m$, and $\delta_m$. The fixed values are set to be identical to the initial values of the learnable Fourier feature mapping, as discussed in Section 4.1.

These experiments are conducted on the "one missing block" setting from Table 1. We use electricity and NP datasets for this evaluation, and the results are summarized in Table 4. As shown in Table 4, using learnable Fourier mapping in NeRT yields better performance compared to using fixed Fourier mapping in both interpolation and extrapolation tasks. That is, by adapting learnable factors to Fourier feature mapping, NeRT successfully learns a set of frequencies that better represents the specific time series.

# D   ADDITIONAL EXPERIMENTS WITH AN ODE-BASED SYNTHETIC TIME SERIES

## D.1   EXPERIMENTAL SETUPS

To evaluate the effectiveness of our proposed NeRT in an ideal setting without any noise, we conduct experiments (cf. Figure 1). We use the damped oscillation ODE, which can represent harmonic and oscillatory motion, as the benchmark dataset. The specific equation and analytic solution are as follows:

$$m_d \frac{d^2 x}{dt^2} + b_d \frac{dx}{dt} + k_d x = 0,$$
$$x(t) = A_d e^{-\frac{b_d}{2m}t} \cos(\omega_d t + \varphi). \tag{4}$$

Equation 4 represents the motion equation that accounts for all the forces acting on the object, where $m_d$ denotes the mass of an object, and $b$ and $k_d$ are physical constants. We set $m_d = 1$, $b_d = 0$ (undamping) or 4 (damping), $\omega_d = 50$, and $A_d = 10$. All experiments employ 1,000 epochs.

## D.2   EXPERIMENTAL RESULTS

In this section, we do additional experiments on undamping/damping oscillatory signals. For each ODEs, we do interpolation, extrapolation, and a mixed task, which do both interpolation and extrapolation at the same time. Since extrapolation task for a damping oscillatory signal is in the main paper, we list results for other five experiments, and the results are summarized in Table 5.

Table 5: Additional results with an ODE-based synthetic time series

|          | Task | SIREN | FFN | NeRT |
|----------|------|-------|-----|------|
| Undamping | Interpolation | $96.6879_{\pm 31.0011}$ | $0.0856_{\pm 0.0445}$ | $\mathbf{0.0183_{\pm 0.0123}}$ |
|          | Extrapolation | $121.8907_{\pm 23.9423}$ | $1.0917_{\pm 0.7321}$ | $\mathbf{0.4109_{\pm 0.4335}}$ |
|          | Interp+Extrap | $93.3860_{\pm 38.3511}$ | $0.1270_{\pm 0.0888}$ | $\mathbf{0.0336_{\pm 0.0204}}$ |
| Damping  | Interpolation | $0.1846_{\pm 0.0796}$ | $0.0017_{\pm 0.0006}$ | $\mathbf{0.0011_{\pm 0.0008}}$ |
|          | Interp+Extrap | $0.2758_{\pm 0.1007}$ | $0.0034_{\pm 0.0018}$ | $\mathbf{0.0004_{\pm 0.0002}}$ |

As shown in Table 5, our proposed NeRT outperforms SIREN and FFN in every task with a big margin. Visualization of the results can be found in Figures 9, 10, 11, 12, 13

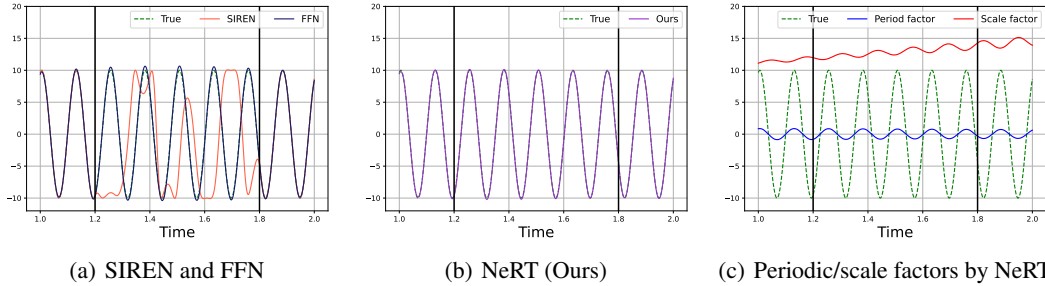

| (a) SIREN and FFN | (b) NeRT (Ours) | (c) Periodic/scale factors by NeRT |

Figure 9: **Interpolation task on an undamping oscillatory signal.** Results of interpolation task with an ODE (Figures 9(a)-(b)) and extracted factors during training (Figure 9(c)). The inside of the two solid lines is a testing range ($t \in [1.2, 1.8]$) and the outside is a training range ($t \in [1.0, 1.2], [1.8, 2.0]$).

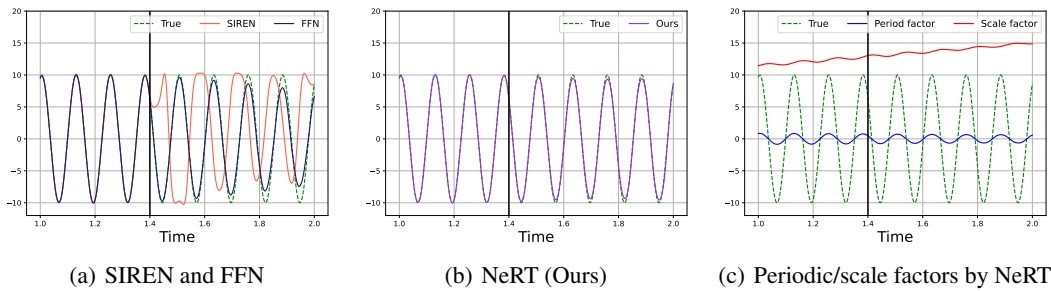

| (a) SIREN and FFN | (b) NeRT (Ours) | (c) Periodic/scale factors by NeRT |

Figure 10: **Extrapolation task on an undamping oscillatory signal.** Results of extrapolation task with an ODE (Figures 10(a)-(b)) and extracted factors during training (Figure 10(c)). The left side of the solid line represents the training range ($t \in [1.0, 1.4]$), while the right side represents the testing range ($t \in [1.4, 2.0]$).

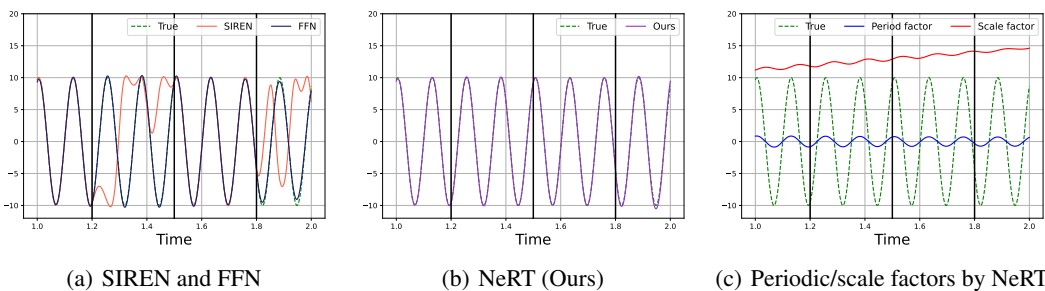

| (a) SIREN and FFN | (b) NeRT (Ours) | (c) Periodic/scale factors by NeRT |

Figure 11: **Interpolation and Extrapolation task on an undamping oscillatory signal.** Results of interpolation and extrapolation tasks with an ODE (Figures 11(a)-(b)) and extracted factors during training (Figure 11(c)). The training range are $t \in [1.0, 1.2], [1.5, 1.8]$, and the testing range are $t \in [1.2, 1.5], [1.8, 2.0]$.

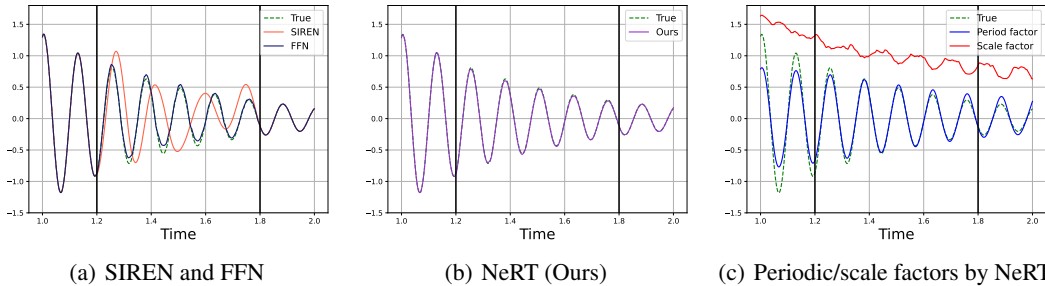

(a) SIREN and FFN          (b) NeRT (Ours)          (c) Periodic/scale factors by NeRT

Figure 12: **Preliminary study with a damping oscillatory signal.** Results of interpolation task with an ODE (Figures 12(a)-(b)) and extracted factors during training (Figure 12(c)). The inside of the two solid lines represents the testing range ($t \in [1.2, 1.8]$, while the outside represents the training range ($t \in [1.0, 1.2], [1.8, 2.0]$).

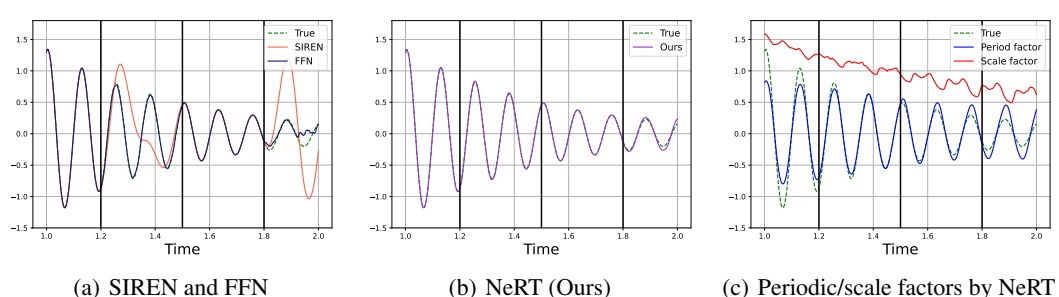

(a) SIREN and FFN          (b) NeRT (Ours)          (c) Periodic/scale factors by NeRT

Figure 13: **Preliminary study with a damping oscillatory signal.** Results of interpolation and extrapolation tasks with an ODE (Figures 13(a)-(b)) and extracted factors during training (Figure 13(c)). The training range are $t \in [1.0, 1.2], [1.5, 1.8]$, and the testing range are $t \in [1.2, 1.5], [1.8, 2.0]$.

# E    ALGORITHM

To provide detailed explanations of the training process of NeRT, we present the following training Algorithm 1.

---

**Algorithm 1** Training of the proposed method

---

/* Training */
**Training datasets:** $M$-dimensional multi-variate time series sequence $\{(\mathbf{x}_i, t_i)\}_{i=1}^{N}$
**Input:** A set of training sampled spatiotemperal coordinate: $\{\mathbf{c}_i^t, (\{\mathbf{c}_j^f\}_{j=1}^{M})\}_{i=1}^{N}$
Initialize the parameters of NeRT $\{\theta_t, \theta_F, \theta_f, \theta_s, \theta_p\}$
**for** $epoch = 1$ to $ep$ **do**
 Compute forward pass: $\hat{x}_{i,j}((\mathbf{c}_i^t, \mathbf{c}_j^f); \{\theta_t, \theta_F, \theta_f, \theta_s, \theta_p\})$
 Compute MSE: $(\hat{x}_{i,j} - x_{i,j})^2$
 Compute backward pass and update the parameters
 Compute the validation error with validation data
**end for**
**Output:** Optimal parameters of NeRT (Lowest validation error)

---

# F   DETAILED DESCRIPTION OF DATASETS

## F.1   2D-HELMHOLTZ EQUATION

The 2D Helmholtz equation, used in Section 5.2, is a differential equation utilized for modeling wave and electromagnetic phenomena. We experiment with the condition of the Equation 5 and are able to directly obtain the analytical solution $u(x, y) = \sin(a_1 \pi x) \sin(a_2 \pi y)$.

## F.2   TIME SERIES DATASETS

Table 6: **Dataset statistics.** Max. length (resp. Min. length) is the longest (resp. shortest) timestamp length among the timestamps of the features in the samples.

| Dataset | Periodic time series | | | | Long-term time series | | |
|---|---|---|---|---|---|---|---|
| | Electricity | Traffic | Caiso | NP | ETTh1 | ETTh2 | National Illness |
| Frequency | hourly | hourly | hourly | hourly | hourly | hourly | weekly |
| Start date | 2012-01-01 | 2008-01-02 | 2013-01-01 | 2013-01-01 | 2016-07-01 | 2016-07-01 | 2002-01-01 |
| End date | 2015-01-01 | 2009-03-31 | 2021-06-30 | 2020-12-31 | 2018-06-26 | 2018-06-26 | 2020-06-30 |
| # features | 1 | 1 | 1 | 1 | 7 | 7 | 7 |
| Max. length | 26,304 | 10,560 | 74,079 | 70,120 | 17,420 | 17,420 | 966 |
| Min. length | 26,271 | 10,512 | 41,547 | 70,076 | 17,420 | 17,420 | 966 |

Experiments on time series datasets consists of two parts: i) periodic time series, and ii) long-term time series. We use four datasets used as benchmark datasets in (Fan et al., 2022) for the periodic time series task and three datasets from (Wen et al., 2022) for the long-term time series task. As shown in table 6, in order to show how NeRT predicts in various scenarios, we choose datasets with a wide range of length and frequency, including both uni-variate and multi-variate datasets. Detailed descriptions of datasets are as follows:

- Periodic time series
    - Electricity contains hourly records of electricity consumption from 2012 to 2014.
    - Traffic consists of hourly data from the sensors in San Francisco freeways, providing information on the road occupancy rates between 2015 and 2016.
    - Caiso comprises hourly actual electricity load series in various zones across California from 2013 to 2021.
    - NP contains a collection of hourly energy production volume from 2013 to 2020 in several European countries.
- Long-term time series
    - ETTh1 and ETTh2 (Zhou et al., 2021) are hourly collected ETT (Electricity Transformer Temperature) datasets from July 2016 to July 2018.
    - National Illness is a weekly collected medical dataset from the Centers for Disease Control and Prevention of the United States. It contains the information of patients with influenza-like illness spanning from 2002 to 2021.

## G  DETAILED EXPERIMENTAL RESULTS ON 2D HELMHOLTZ EQUATION

The 2D Helmhotz equations are used as a benchmark problem in (McClenny & Braga-Neto, 2020). The form of the PDE is as follows:

$$\frac{\partial^2 u(x,y)}{\partial x^2} + \frac{\partial^2 u(x,y)}{\partial y^2} + k^2 u(x,y) - q(x,y) = 0 \tag{5}$$

$$q(x,y) = (-(a_1\pi)^2 - (a_2\pi)^2 + k^2)\sin(a_1\pi x)\sin(a_2\pi y)$$

where $k$ is a constant, and $q(x,y)$ is a source term. NeRT and baselines are trained to predict the solution $u$ at a given location $(x,y)$. Unlike the multi-variate time series, $u$ is uni-variate, so NeRT uses only $\psi_t$ as its encoder — we feed the raw coordinate $(x,y)$ instead of $\mathbf{c}_i^t$.

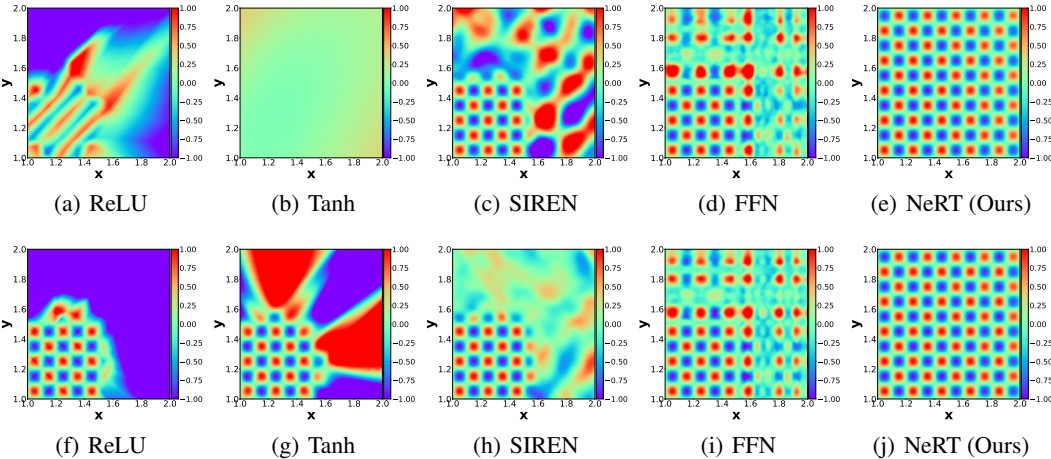

(a) ReLU  (b) Tanh  (c) SIREN  (d) FFN  (e) NeRT (Ours)

(f) ReLU  (g) Tanh  (h) SIREN  (i) FFN  (j) NeRT (Ours)

Figure 14: Detailed experimental results

Table 7: Experimental resultes of Helmholtz equations

| Epoch | ReLU | Tanh | SIREN | FFN | NeRT |
|-------|------|------|-------|-----|------|
| 1,000 | $1.8735\pm_{1.1729}$ | $0.2966\pm_{0.0719}$ | $0.8535\pm_{0.2780}$ | $0.3456\pm_{0.0187}$ | $\mathbf{0.0042\pm_{0.0010}}$ |
| 10,000 | $52.1329\pm_{0.0719}$ | $3.7004\pm_{2.5366}$ | $0.3260\pm_{0.0114}$ | $0.3109\pm_{0.0173}$ | $\mathbf{0.0014\pm_{0.0008}}$ |

In the extension of Experiment 5.2, to provide more comprehensive analysis, we increase the number of epochs and add a baseline. Figures 14(a)-(e) represent the results after training for 1,000 epochs, while Figures 14(f)-(g) depict the results after training for 10,000 epochs. Furthermore, keeping all hyperparameters and model sizes the same, we add baselines by changing the activation function to ReLU and Tanh. This is depicted in the first and second columns of Figure 14. As shown in Figures 14(a) and (b), models using ReLU and Tanh activation functions struggle even to learn the training range. On the other hand, all INR models demonstrate successfully precise learning of the training range ($x \in [1.0, 1.5], y \in [1.0, 1.5]$) within just 1,000 epoch. When training is extended to 10,000 epochs, all models managed to learn the training range, but only NeRT successfully predict the test range ($x \in [1.5, 2.0], y \in [1.5, 2.0]$).

# H    EXPERIMENTS ON PERIODIC TIME SERIES

## H.1    DETAILED EXPERIMENTAL SETUPS

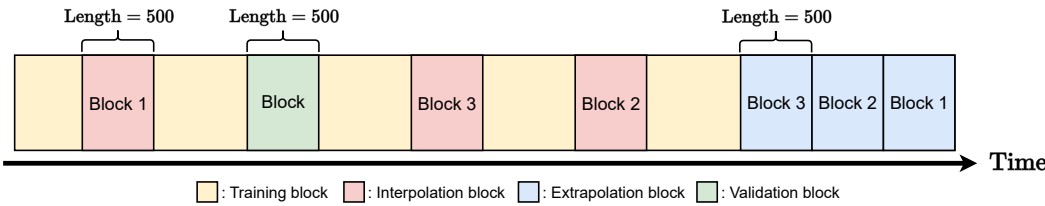

Figure 15: Experimental setup on periodic time series.

Table 8: Best hyperparameter configurations in the periodic time series task

|  | $\omega^{\text{init}}$ | $\omega^{\text{inner}}$ | $\dim(\psi_t)$ | $\dim(\psi_F)$ | $\dim(\mathbf{h}_p)$ | $\dim(\mathbf{h}_s)$ |
|---|---|---|---|---|---|---|
| Electricity | 5.0 | 1.0 | 30 | 30 | 10 | 30 |
| Traffic | 10.0 | 1.0 | 30 | 10 | 50 | 10 |
| Caiso | 10.0 | 1.0 | 10 | 30 | 50 | 10 |
| NP | 3.0 | 3.0 | 30 | 30 | 10 | 30 |

We design the experiment using the first 10 samples, each of which consists of 12 blocks (cf. Figure 15), from each dataset and conduct an experiment to fill in the values of missing blocks. In a sample, we perform both the interpolation and the extrapolation tasks. Detailed locations and constructions are summarized in Figure 15. As shown in Figure 15, in a sample there are three interpolation blocks colored in red, three extrapolation blocks colored in blue, one validation block colored in green, and the remaining yellow parts represent the training dataset. Each block has a length of 500, and the missing blocks for each task, as specified as "# blocks" in Table 1, are as follows:

- # blocks = 1 means that we perform the interpolation and extrapolation tasks for Block 1.

- # blocks = 2 means that we perform the interpolation and extrapolation tasks for Block 1 and Block 2.

- # blocks = 3 means that we perform the interpolation and extrapolation tasks for Block 1, Block 2 and Block 3.

A validation block is not used for training in every task, and used for the purpose of validation. Each model is trained for 2,000 epochs.

For hyperparameters, we set $S_{max}$ to 1 and for the fair comparison, we use similar model sizes for all methods and share the frequencies across the models. There are two frequencies used as hyperparameters, $\omega^{\text{init}}$ and $\omega^{\text{inner}}$. $\omega^{\text{init}}$ is used in our learnable Fourier feature mapping and corresponds to $b$ in Equation 2. $\omega^{\text{inner}}$ denotes the frequency of the sinusoidal function $\rho_s$ inn Equation 3. For the number of layers, we set $L_t$, $L_f$ and $L_s$ to 2, and $L_p$ to 5. The best hyperparameter configurations of NeRT in the periodic time series task are summarized in Table 8. We use an additional FC layer after Equation 2 and let $\mathbf{h}_p$ and $\mathbf{h}_s$ be the hidden vectors of the period and scale decoders, respectively.

## H.2    ADDITIONAL EXPERIMENTAL RESULTS

We conduct periodic time series experiments on two numerical methods, Linear (linear interpolation) and Cubic (cubic interpolation), and report the results in Table 9. As shown in Table 9, NeRT beats two numerical methods. We compare NeRT to Linear and Cubic only for the interpolation task, since those numerical methods are not able to extrapolate.

Additionally, in Figures 16, 17, 18, and 19, we show the visualizations of the remaining datasets' interpolation and extrapolation results that are not included in the main paper. In Figures 16 and 18, we show the results at the best epoch, i.e., the lowest validation error, and in Figures 17 and 19, we show results at the last epoch, i.e., after 2,000 epochs. NeRT avoids overfitting to training data in both cases while other two baselines are significantly overfitted to training data and fail in the testing range.

In Figure 20, we propose how the periodic and scale factors of NeRT work in the periodic time series interpolation and extrapolation tasks. Figures 20 (a)-(d) correspond to the interpolation task and Figures 20 (e)-(h) correspond to the extrapolation task.

Table 9: **Additional experimental results on periodic time series.** The best results are reported in boldface.

| Dataset | # blocks | Interpolation | | |
|---|---|---|---|---|
| | | Linear | Cubic | NeRT |
| Electricity | 1 | 0.0126 | 61.4654 | $\mathbf{0.0061}\pm\mathbf{0.0006}$ |
| | 2 | 0.0155 | 47.4215 | $\mathbf{0.0057}\pm\mathbf{0.0005}$ |
| | 3 | 0.0182 | 41.9424 | $\mathbf{0.0056}\pm\mathbf{0.0006}$ |
| Traffic | 1 | 0.0220 | 4.4286 | $\mathbf{0.0057}\pm\mathbf{0.0002}$ |
| | 2 | 0.0191 | 3.0941 | $\mathbf{0.0062}\pm\mathbf{0.0000}$ |
| | 3 | 0.0191 | 4.1447 | $\mathbf{0.0055}\pm\mathbf{0.0002}$ |
| Caiso | 1 | 0.0223 | 8.8545 | $\mathbf{0.0047}\pm\mathbf{0.0012}$ |
| | 2 | 0.0171 | 6.3525 | $\mathbf{0.0041}\pm\mathbf{0.0002}$ |
| | 3 | 0.0171 | 6.4894 | $\mathbf{0.0099}\pm\mathbf{0.0007}$ |
| NP | 1 | 0.0426 | 1.3798 | $\mathbf{0.0149}\pm\mathbf{0.0005}$ |
| | 2 | 0.0477 | 2.2489 | $\mathbf{0.0186}\pm\mathbf{0.0008}$ |
| | 3 | 0.0454 | 3.8557 | $\mathbf{0.0196}\pm\mathbf{0.0006}$ |

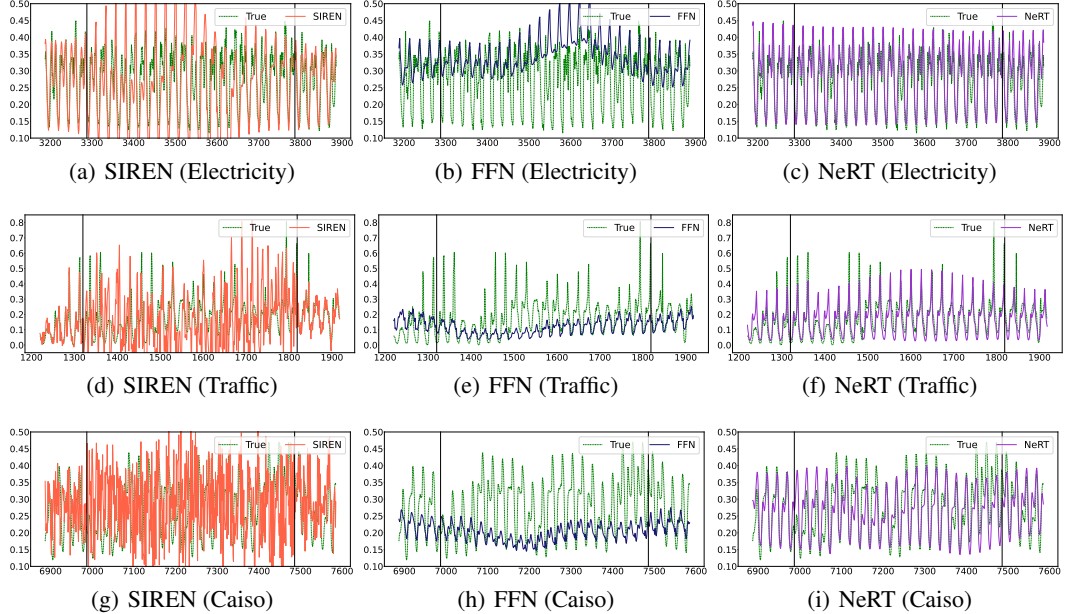

Figure 16: **Interpolation results on missing intervals at the lowest validation error checkpoint.**
Interpolation results in Electricity (Figures 16 (a)-(c)), in Traffic (Figures 16 (d)-(f)), and in Caiso
(Figures 16 (g)-(i)). The space between the two solid lines represents the testing range, while the
outer two parts represent the training range.

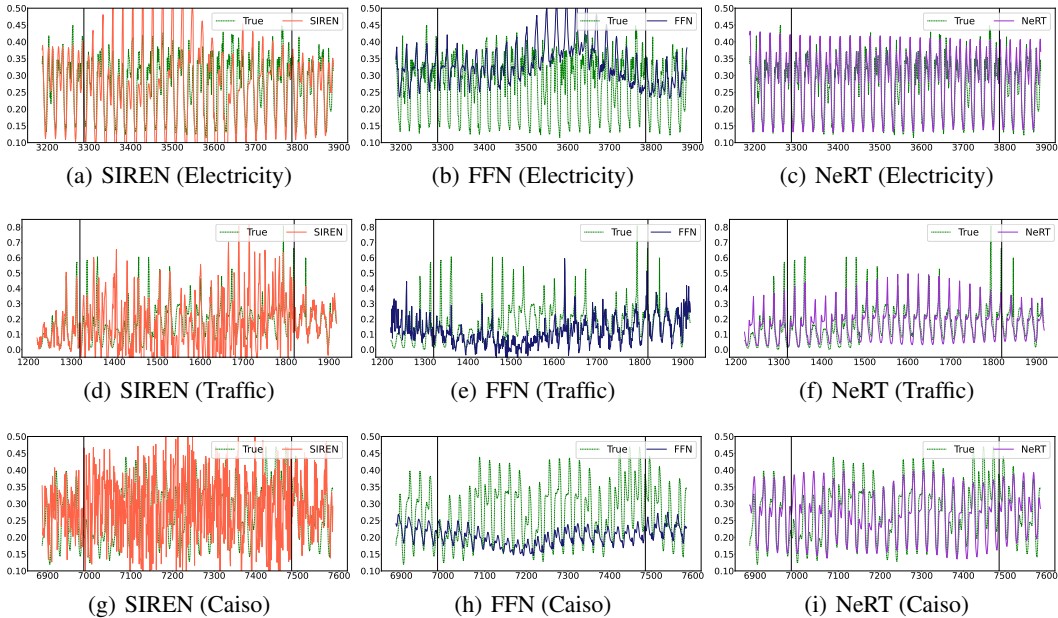

Figure 17: **Interpolation results on missing intervals after the last epoch.** Interpolation results in
Electricity (Figures 17 (a)-(c)), in Traffic (Figures 17 (d)-(f)), and in Caiso (Figures 17 (g)-(i)). The
space between the two solid lines represents the testing range, while the outer two parts represent the
training range.

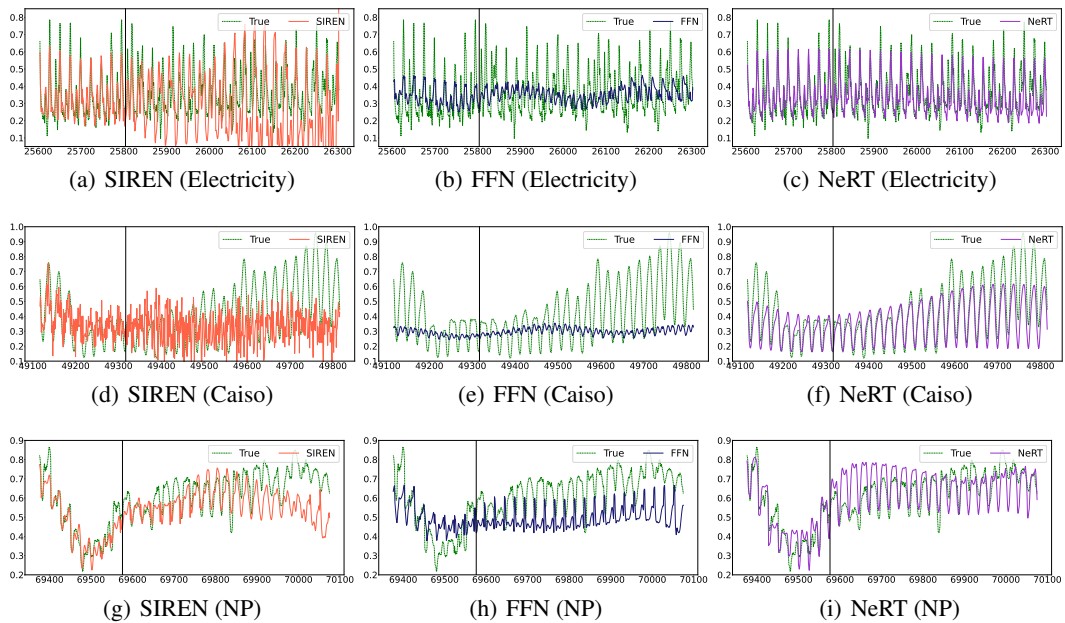

Figure 18: **Extrapolation results on missing intervals at the lowest validation error checkpoint.** Extrapolation results in Electricity (Figures 18 (a)-(c)), in Caiso (Figures 18 (d)-(f)), and in NP (Figures 18 (g)-(i)). The right area after the solid vertical line is a testing range.

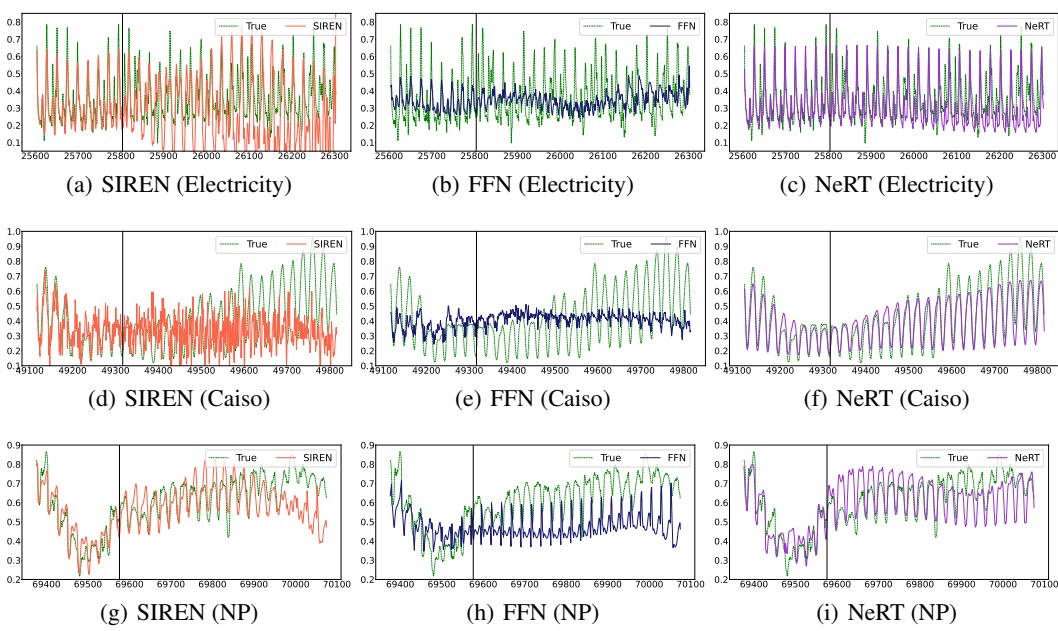

Figure 19: **Extrapolation results on missing intervals after the last epoch.** Extrapolation results in Electricity (Figures 19 (a)-(c)), in Caiso (Figures 19 (d)-(f)), and in NP (Figures 19 (g)-(i)). The right area after the solid vertical line is a testing range.

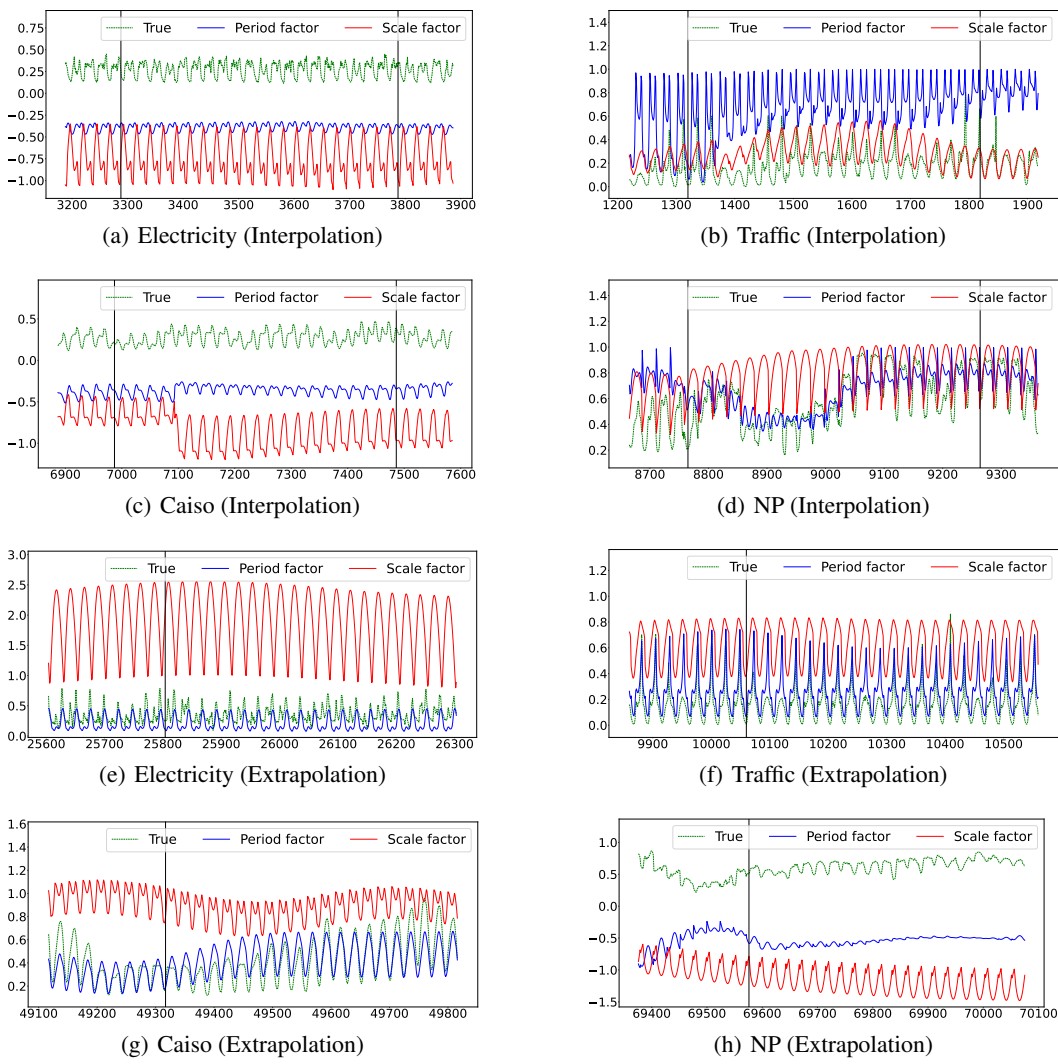

Figure 20: **Periodic and Scale factors trained on periodic time series.** Interpolation results (Figures 20 (a)-(d)), where the space between the two solid lines represents the testing range, while the outer two parts represent the training range, and extrapolation results (Figures 20 (e)-(h)), where right area after the solid vertical line is a testing range.

# I ADDITIONAL COMPARISON WITH TIME-SERIES BASELINES

## I.1 EXPERIMENTAL SETUPS

In this section, we conduct comprehensive analyses between our model and existing time-series baselines to support the necessity of adapting INR specifically to time series data (cf. Section 2). We use four datasets used in Section 5.1, and we have meticulously arranged the experimental setup in the subsequent manner for the purpose of this study. To be specific, we compare the forecasting performance by varying the input window size $m$ of $\{48, 96, 192\}$ and the output window size $n$ of $\{96, 192, 336, 720\}$, following the overall experimental configuration of (Zeng et al., 2022). For each data sample, we fix the total length to 2,880 with a train size of 1,440 and a validation and test sizes of 720. To assess the model performance, we use mean-squared error (MSE) of the test range at the epoch where the best MSE on the validation range is achieved.

**Baselines** To evaluate the performance of NeRT, we compare it with eight existing time-series baselines. As representatives of traditional time series models, we use RNN and LSTM as baselines. Additionally, we compare NeRT against models in the long-term time series forecasting domain, including Linear-based models, i.e., Linear, DLinear, and NLinear from (Zeng et al., 2022), and Transformer-based baselines, i.e., Autoformer (Wu et al., 2021), Informer (Zhou et al., 2021), and FEDformer (Zhou et al., 2022). Furthermore, we employ Latent ODE (Rubanova et al., 2019) and Neural CDE (Kidger et al., 2020) as Neural ODE-based models.

**Training methodological differences between NeRT and other baselines** Existing time-series baselines are highly sensitive to varying input and output window sizes (cf. **L2** of Section 2)), so they need repeated experiments for each window size change. In contrast, since our NeRT operates independently of window size, we need to train NeRT only once for each dataset. During the single training, NeRT makes a single set of predictions for all 720 points, and then calculate MSEs for all the output window sizes of 96, 192, 336, and 720, respectively.

## I.2 EXPERIMENTAL RESULTS

We summarize experimental results of time series forecasting depending on the input/output window sizes in Tables 10, 11, and 12. Note that while other time series baselines are trained individually for each combination of the input/output window sizes, NeRT employs only a single model for each dataset. As shown in the tables, NeRT consistently shows the best MSE regardless of the dataset and window size. On top of that, the results of the baselines are highly affected by the input/output window sizes (cf. Appendix B), making hard to choose an appropriate window size setting. Figure 21 depicts how MSE changes as $n$ varies, with $m$ fixed to 96 and 192. In every case, NeRT shows the lowest MSE with the lowest slope, compared to other baselines. Additionally, Figure 22 shows how the models predict $n$ values given the input window size $m$ on the Traffic dataset, where $n = 96$ and $m = 48$ in this setting.

**Computational cost** In Table 13, we describe models' computational complexity in terms of time and memory (cf. **L4** of Section 2). We report the complexity for all window combinations of the baselines individually, whereas with the NeRT's capacity to provide results for all combinations at once using a single model, we report the training complexity of the single NeRT model. The results in Table 13 correspond to an input window size of 96, and each value represents the complexity required during the training of a single data sample. For NeRT, we record the average time/memory complexity needed to train a single data sample, with the values in parentheses indicating the total complexity required for training a single model.

As shown in Table 13, the total memory complexity of NeRT is notably smaller than the memory complexity of all RNN/Transformer/Neural ODE-based models when training just one data sample. This reduction in complexity ranges from being 5.5 times smaller to as much as 1,730 times smaller and in certain instances, it is even smaller than those of Linear-based models. For the time complexity, NeRT outperforms Transformer/Neural ODE-based models, but it is slower than Linear/RNN-based models. In summary, considering the forecasting results from Tables 10, 11, and 12, NeRT demonstrates significant forecasting performance improvements, approximately 2 to 5 times better

than existing time-series models, while still maintaining a sufficiently fast training time complexity and memory complexity similar to Linear-based models.

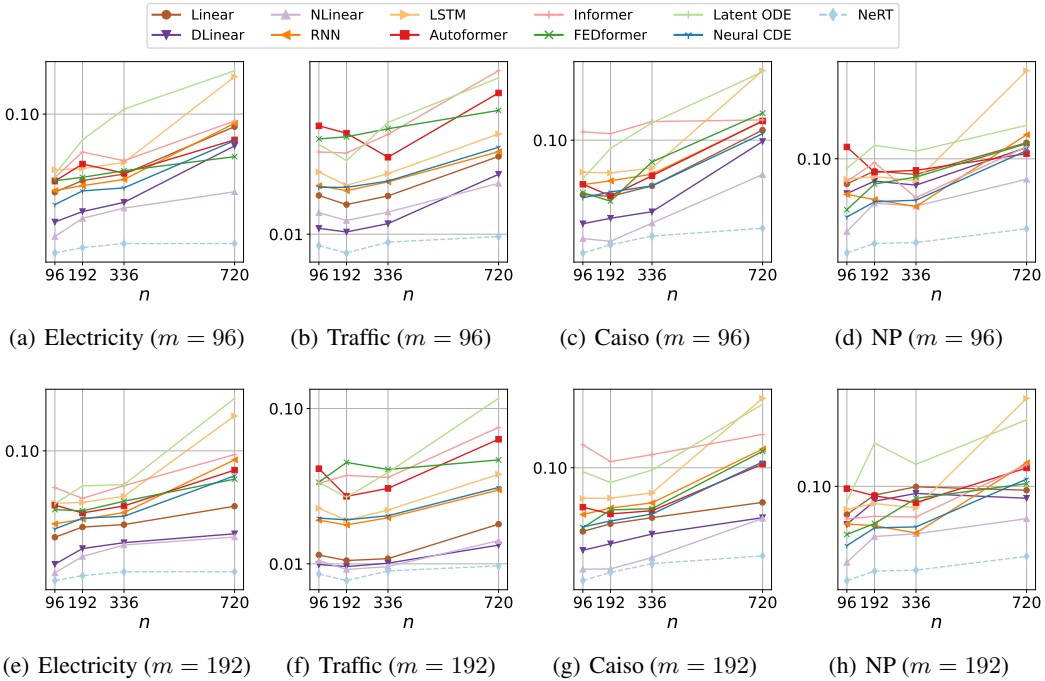

(a) Electricity ($m = 96$)   (b) Traffic ($m = 96$)   (c) Caiso ($m = 96$)   (d) NP ($m = 96$)

(e) Electricity ($m = 192$)   (f) Traffic ($m = 192$)   (g) Caiso ($m = 192$)   (h) NP ($m = 192$)

Figure 21: Comparisons with time-series baselines for varying $n = \{96, 192, 336, 720\}$ (output/prediction window size) with a fixed input window size $m = 96$ ((a)-(d)) and $m = 192$ ((e)-(h)).

Table 10: Comparision with time-series baselines ($m = 48$)

| Dataset | $n$ | Linear-based | | | RNN-based | | Transformer-based | | | Neural ODE-based | | INR-based |
|---|---|---|---|---|---|---|---|---|---|---|---|---|
| | | Linear | DLinear | NLinear | RNN | LSTM | Autoformer | Informer | FEDformer | Latent ODE | Neural CDE | NeRT (Ours) |
| Electricity | 96 | 0.0671 | 0.0556 | 0.0329 | 0.0387 | 0.0495 | 0.0555 | 0.0485 | 0.0387 | 0.0673 | 0.0333 | **0.0174** |
| | 192 | 0.0747 | 0.0629 | 0.0351 | 0.0411 | 0.0514 | 0.0544 | 0.0610 | 0.0392 | 0.0856 | 0.0390 | **0.0186** |
| | 336 | 0.0843 | 0.0689 | 0.0387 | 0.0444 | 0.0547 | 0.0520 | 0.0641 | 0.0460 | 0.1500 | 0.0403 | **0.0196** |
| | 720 | 0.1461 | 0.1326 | 0.0482 | 0.0890 | 0.1601 | 0.0741 | 0.1003 | 0.0668 | 0.1565 | 0.0720 | **0.0196** |
| Traffic | 96 | 0.0278 | 0.0240 | 0.0215 | 0.0191 | 0.0229 | 0.0395 | 0.0412 | 0.0290 | 0.0349 | 0.0192 | **0.0086** |
| | 192 | 0.0275 | 0.0225 | 0.0202 | 0.0179 | 0.0192 | 0.0368 | 0.0362 | 0.0329 | 0.0528 | 0.0194 | **0.0078** |
| | 336 | 0.0303 | 0.0249 | 0.0219 | 0.0201 | 0.0224 | 0.0325 | 0.0410 | 0.0315 | 0.0460 | 0.0208 | **0.0090** |
| | 720 | 0.0446 | 0.0418 | 0.0289 | 0.0297 | 0.0377 | 0.0652 | 0.0670 | 0.0439 | 0.1036 | 0.0320 | **0.0097** |
| Caiso | 96 | 0.0989 | 0.0843 | 0.0448 | 0.0556 | 0.0671 | 0.0551 | 0.0917 | 0.0455 | 0.1088 | 0.0501 | **0.0236** |
| | 192 | 0.0972 | 0.0799 | 0.0450 | 0.0607 | 0.0647 | 0.0539 | 0.1322 | 0.0410 | 0.1538 | 0.0533 | **0.0262** |
| | 336 | 0.1154 | 0.0927 | 0.0548 | 0.0658 | 0.0711 | 0.0683 | 0.0824 | 0.0602 | 0.1841 | 0.0573 | **0.0293** |
| | 720 | 0.2106 | 0.1948 | 0.0808 | 0.1276 | 0.2431 | 0.1471 | 0.1876 | 0.1467 | 0.3257 | 0.1103 | **0.0324** |
| NP | 96 | 0.1114 | 0.1085 | 0.0690 | 0.0697 | 0.0790 | 0.0784 | 0.0703 | 0.0598 | 0.1060 | 0.0562 | **0.0370** |
| | 192 | 0.1007 | 0.0980 | 0.0795 | 0.0649 | 0.0814 | 0.0733 | 0.0803 | 0.0747 | 0.1522 | 0.0562 | **0.0409** |
| | 336 | 0.1072 | 0.1035 | 0.0932 | 0.0606 | 0.0794 | 0.0860 | 0.0727 | 0.0849 | 0.1465 | 0.0669 | **0.0413** |
| | 720 | 0.2223 | 0.2217 | 0.1414 | 0.1292 | 0.2534 | 0.1073 | 0.0984 | 0.0820 | 0.3414 | 0.1120 | **0.0478** |

Table 11: Comparision with time-series baselines ($m = 96$)

| Dataset | $n$ | Linear-based | | | RNN-based | | Transformer-based | | | Neural ODE-based | | INR-based |
| | | Linear | DLinear | NLinear | RNN | LSTM | Autoformer | Informer | FEDformer | Latent ODE | Neural CDE | NeRT (Ours) |
|---|---|---|---|---|---|---|---|---|---|---|---|---|
| Electricity | 96 | 0.0374 | 0.0257 | 0.0214 | 0.0382 | 0.0496 | 0.0430 | 0.0450 | 0.0432 | 0.0468 | 0.0320 | **0.0174** |
| | 192 | 0.0433 | 0.0293 | 0.0269 | 0.0406 | 0.0503 | 0.0533 | 0.0620 | 0.0451 | 0.0723 | 0.0380 | **0.0186** |
| | 336 | 0.0470 | 0.0329 | 0.0306 | 0.0439 | 0.0544 | 0.0476 | 0.0556 | 0.0490 | 0.1064 | 0.0395 | **0.0196** |
| | 720 | 0.0853 | 0.0670 | 0.0376 | 0.0890 | 0.1601 | 0.0723 | 0.0917 | 0.0585 | 0.1726 | 0.0716 | **0.0196** |
| Traffic | 96 | 0.0167 | 0.0108 | 0.0133 | 0.0190 | 0.0228 | 0.0420 | 0.0297 | 0.0353 | 0.0328 | 0.0186 | **0.0086** |
| | 192 | 0.0148 | 0.0103 | 0.0120 | 0.0178 | 0.0191 | 0.0381 | 0.0293 | 0.0363 | 0.0265 | 0.0186 | **0.0078** |
| | 336 | 0.0166 | 0.0115 | 0.0134 | 0.0200 | 0.0223 | 0.0277 | 0.0375 | 0.0404 | 0.0439 | 0.0202 | **0.0090** |
| | 720 | 0.0280 | 0.0221 | 0.0196 | 0.0298 | 0.0377 | 0.0649 | 0.0871 | 0.0515 | 0.0792 | 0.0316 | **0.0097** |
| Caiso | 96 | 0.0498 | 0.0343 | 0.0284 | 0.0566 | 0.0663 | 0.0569 | 0.1112 | 0.0511 | 0.0611 | 0.0478 | **0.0236** |
| | 192 | 0.0487 | 0.0368 | 0.0274 | 0.0594 | 0.0658 | 0.0490 | 0.1086 | 0.0458 | 0.0900 | 0.0515 | **0.0262** |
| | 336 | 0.0557 | 0.0400 | 0.0346 | 0.0647 | 0.0700 | 0.0634 | 0.1268 | 0.0757 | 0.1249 | 0.0557 | **0.0293** |
| | 720 | 0.1138 | 0.0983 | 0.0644 | 0.1273 | 0.2432 | 0.1279 | 0.1296 | 0.1413 | 0.2397 | 0.1086 | **0.0324** |
| NP | 96 | 0.0767 | 0.0693 | 0.0464 | 0.0687 | 0.0807 | 0.1133 | 0.0787 | 0.0584 | 0.0903 | 0.0541 | **0.0370** |
| | 192 | 0.0879 | 0.0790 | 0.0626 | 0.0650 | 0.0828 | 0.0868 | 0.0961 | 0.0766 | 0.1152 | 0.0635 | **0.0409** |
| | 336 | 0.0847 | 0.0757 | 0.0607 | 0.0604 | 0.0794 | 0.0885 | 0.0663 | 0.0825 | 0.1084 | 0.0646 | **0.0413** |
| | 720 | 0.1185 | 0.1126 | 0.0806 | 0.1290 | 0.2534 | 0.1056 | 0.1136 | 0.1173 | 0.1424 | 0.1090 | **0.0478** |

Table 12: Comparision with time-series baselines ($m = 192$)

| Dataset | $n$ | Linear-based | | | RNN-based | | Transformer-based | | | Neural ODE-based | | INR-based |
| | | Linear | DLinear | NLinear | RNN | LSTM | Autoformer | Informer | FEDformer | Latent ODE | Neural CDE | NeRT (Ours) |
|---|---|---|---|---|---|---|---|---|---|---|---|---|
| Electricity | 96 | 0.0312 | 0.0217 | 0.0193 | 0.0374 | 0.0493 | 0.0481 | 0.0609 | 0.0453 | 0.0489 | 0.0349 | **0.0174** |
| | 192 | 0.0358 | 0.0268 | 0.0241 | 0.0398 | 0.0499 | 0.0433 | 0.0527 | 0.0445 | 0.0624 | 0.0403 | **0.0186** |
| | 336 | 0.0370 | 0.0290 | 0.0281 | 0.0435 | 0.0542 | 0.0476 | 0.0626 | 0.0507 | 0.0634 | 0.0414 | **0.0196** |
| | 720 | 0.0474 | 0.0327 | 0.0313 | 0.0886 | 0.1602 | 0.0771 | 0.0951 | 0.0683 | 0.2029 | 0.0714 | **0.0196** |
| Traffic | 96 | 0.0114 | 0.0099 | 0.0105 | 0.0191 | 0.0228 | 0.0410 | 0.0330 | 0.0335 | 0.0335 | 0.0197 | **0.0086** |
| | 192 | 0.0105 | 0.0096 | 0.0092 | 0.0178 | 0.0191 | 0.0272 | 0.0371 | 0.0450 | 0.0268 | 0.0192 | **0.0078** |
| | 336 | 0.0108 | 0.0101 | 0.0096 | 0.0198 | 0.0222 | 0.0306 | 0.0360 | 0.0405 | 0.0387 | 0.0204 | **0.0090** |
| | 720 | 0.0180 | 0.0132 | 0.0141 | 0.0299 | 0.0377 | 0.0634 | 0.0756 | 0.0467 | 0.1163 | 0.0310 | **0.0097** |
| Caiso | 96 | 0.0443 | 0.0348 | 0.0273 | 0.0550 | 0.0676 | 0.0604 | 0.1342 | 0.0464 | 0.0951 | 0.0469 | **0.0236** |
| | 192 | 0.0488 | 0.0378 | 0.0274 | 0.0597 | 0.0678 | 0.0554 | 0.1082 | 0.0583 | 0.0828 | 0.0506 | **0.0262** |
| | 336 | 0.0528 | 0.0428 | 0.0317 | 0.0638 | 0.0725 | 0.0577 | 0.1183 | 0.0592 | 0.0971 | 0.0553 | **0.0293** |
| | 720 | 0.0642 | 0.0528 | 0.0523 | 0.1270 | 0.2431 | 0.1048 | 0.1533 | 0.1234 | 0.2253 | 0.1077 | **0.0324** |
| NP | 96 | 0.0743 | 0.0670 | 0.0448 | 0.0672 | 0.0786 | 0.0976 | 0.0709 | 0.0602 | 0.0852 | 0.0535 | **0.0370** |
| | 192 | 0.0912 | 0.0855 | 0.0589 | 0.0661 | 0.0834 | 0.0905 | 0.0729 | 0.0675 | 0.1579 | 0.0645 | **0.0409** |
| | 336 | 0.0995 | 0.0928 | 0.0606 | 0.0611 | 0.0797 | 0.0842 | 0.0724 | 0.0881 | 0.1261 | 0.0652 | **0.0413** |
| | 720 | 0.0960 | 0.0882 | 0.0711 | 0.1282 | 0.2530 | 0.1214 | 0.1254 | 0.1029 | 0.2017 | 0.1079 | **0.0478** |

Table 13: Computational cost with $m$ at 96. Each value is measured during training one data sample. For NeRT, since it is not trained for each window combination, we report the average cost required for training one sample, with total amount in the parentheses.

| Complexity | $n$ | Linear-based | | | RNN-based | | Transformer-based | | | Neural ODE-based | | INR-based |
| | | Linear | DLinear | NLinear | RNN | LSTM | Autoformer | Informer | FEDformer | Latent ODE | Neural CDE | NeRT |
|---|---|---|---|---|---|---|---|---|---|---|---|---|
| Time (sec) | 96 | 0.0333 | 0.0449 | 0.0353 | 0.0864 | 0.0951 | 0.9343 | 0.8525 | 9.3507 | 15.0144 | 9.9457 | 0.2345 |
| | 192 | 0.0329 | 0.0446 | 0.0347 | 0.0886 | 0.0929 | 1.0673 | 0.7802 | 9.4309 | 17.2511 | 9.9699 | (2.8151) |
| | 336 | 0.0320 | 0.0443 | 0.0338 | 0.0898 | 0.0930 | 1.2836 | 0.9100 | 10.7392 | 20.3422 | 9.8612 | |
| | 720 | 0.0202 | 0.0275 | 0.0226 | 0.0335 | 0.0387 | 0.9745 | 0.7102 | 7.7706 | 9.6910 | 5.7095 | |
| Memory (MB) | 96 | 1.0757 | 1.1816 | 1.0991 | 54.2866 | 92.9487 | 888.9063 | 504.9854 | 2254.0459 | 24.2246 | 9.6597 | 0.1318 |
| | 192 | 1.1519 | 1.3169 | 1.1753 | 54.3799 | 93.0425 | 1292.9297 | 679.2822 | 2252.9355 | 26.4951 | 9.6597 | (1.5820) |
| | 336 | 1.2769 | 1.5303 | 1.3003 | 54.5508 | 93.2148 | 1768.7114 | 908.2749 | 2890.7788 | 29.9116 | 9.9468 | |
| | 720 | 1.4126 | 1.8643 | 1.4292 | 46.4707 | 78.8481 | 2735.4766 | 1273.8496 | 3913.7681 | 33.2192 | 8.7642 | |

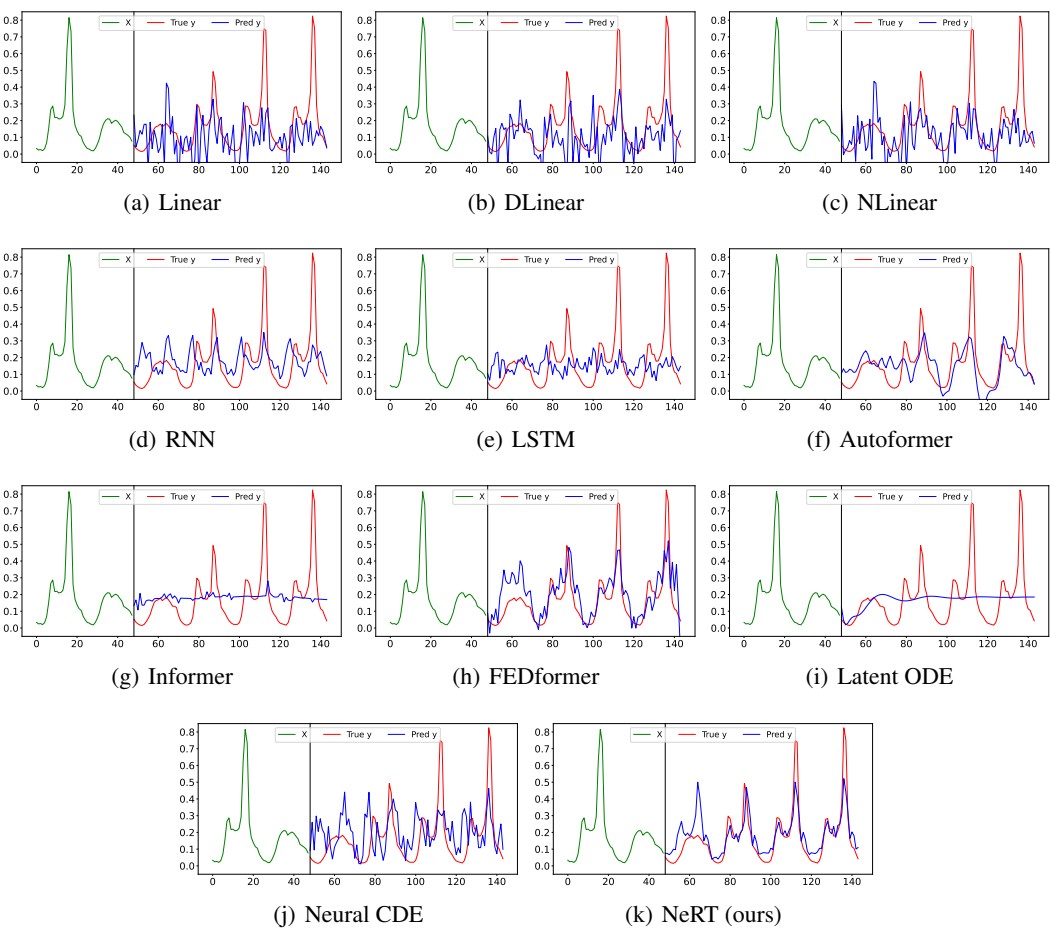

Figure 22: **Forecasting task on Traffic.** We set $m$ to 48 and $n$ to 96. The left side of the solid line represents the input window, while the right side represents the output window.

# J    ADDITIONAL COMPARISON WITH MODULATED INR ON UNSEEN SAMPLES

Table 14: Comparison with Modulated INRs

| Dataset | Task | Modulated SIREN | Modulated FFN | Modulated NeRT (scale) | Modulated NeRT (scale and period) |
|---------|------|-----------------|---------------|------------------------|-----------------------------------|
| Electricity | Imputation | 0.1840 | 0.0355 | 0.0322 | **0.0315** |
| | Forecasting | 0.1844 | 0.0188 | 0.0161 | **0.0149** |
| Traffic | Imputation | 0.0457 | 0.0067 | **0.0009** | 0.0040 |
| | Forecasting | 0.0373 | 0.0116 | **0.0042** | 0.0091 |
| Caiso | Imputation | 0.0306 | 0.0390 | 0.0375 | **0.0264** |
| | Forecasting | 0.0869 | 0.0907 | 0.0663 | **0.0629** |
| NP | Imputation | 0.0361 | 0.0257 | **0.0224** | 0.0229 |
| | Forecasting | 0.0238 | 0.0449 | 0.0441 | **0.0166** |

## J.1    LATENT MODULATION

Fundamentally, a single INR model tends to overfit to a single data sample, making it challenging to represent unseen data samples effectively. To overcome this limitation, recently developed modulation techniques involve i) sharing model parameters and ii) learning sample-specific parameters, enabling INR models capable of representing various data samples. In particular, the latent modulation, introduced in (Dupont et al., 2022), is one of the most effective training methods for INRs to infer unseen samples after learning multiple samples — we strictly follow this training method in this subsection. It is a meta-learning-based modulation approach that allows the representation of diverse data samples by adding an additional learnable bias to each shared MLP layer and for each sample, the biases in all the MLP layers are changed — at the end, all data samples can be somehow learned by the combination of the shared MLP parameters and the sample-specific additional biases. By adopting this concept to all INR-based models used in the paper, which are SIREN, FFN, and NeRT, they are able to predict values of unseen samples.

## J.2    EXPERIMENTAL SETUPS

To ensure a fair comparison, NeRT, SIREN, and FFN all employ the same latent modulation approach. The baselines, referred to as modulated SIREN and modulated FFN, are SIREN and FFM models with latent modulation applied to all layers except the first and last. In the case of modulated NeRT, we propose two variants. Firstly, latent modulation is applied to both the scale decoder and the periodic decoder of vanilla NeRT, denoted Modulated NeRT (scale). Secondly, we apply latent modulation exclusively to the scale decoder, denoted Modulated NeRT (scale and period).

The datasets used in the experiments are the periodic time-series datasets discussed in Section 5.1, and the experiments are conducted in an environment identical to that described in Appendix H. The dimensionality of the modulation vector remains consistent at 256 throughout the training of all models. Additionally, testing is carried out on unseen block of unseen samples that are not part of the training process. Both imputation and forecasting tasks are simultaneously inferred within a single model.

## J.3    EXPERIMENTAL RESULTS

All experimental results are summarized in Table 14, and it can be observed that modulated NeRTs outperform the modulated INR baselines significantly across all benchmark datasets. Particularly, for Traffic dataset, Modulated NeRT (scale) exhibits MSE values that are approximately one-seventh the magnitude of the baseline for interpolation and half the magnitude for extrapolation. Consequently, NeRT shows its scalability to unseen samples with commendable performance.

## K   EXPERIMENTS ON LONG-TERM TIME SERIES

### K.1   DETAILED EXPERIMENTAL SETUPS

Table 15: Hyperparameters of long-term time series

|  | $S_{max}$ | Drop ratio | $\omega^{\text{init}}$ | $\omega^{\text{inner}}$ | $\dim(\psi_t)$ | $\dim(\psi_F)$ | $\dim(\mathbf{h}_p)$ | $\dim(\mathbf{h}_s)$ |
|---|---|---|---|---|---|---|---|---|
| ETTh1 | 100 | 30% | 5.0 | 1.0 | 50 | 30 | 200 | 10 |
|  |  | 50% | 5.0 | 3.0 | 30 | 30 | 100 | 50 |
|  |  | 70% | 10.0 | 3.0 | 30 | 30 | 100 | 30 |
| ETTh2 | 100 | 30% | 5.0 | 1.0 | 10 | 30 | 200 | 10 |
|  |  | 50% | 5.0 | 3.0 | 30 | 30 | 100 | 50 |
|  |  | 70% | 10.0 | 3.0 | 10 | 30 | 50 | 30 |
| National Illness | 1 | 30% | 5.0 | 1.0 | 50 | 10 | 100 | 10 |
|  |  | 50% | 5.0 | 3.0 | 30 | 50 | 30 | 10 |
|  |  | 70% | 10.0 | 3.0 | 10 | 10 | 10 | 10 |

For fair comparison, we share $\omega^{\text{init}}$ and $\omega^{\text{inner}}$ and employ similar model sizes and across the tested models. We note hyperparameter configurations used in long-term time series experiments in Table 15. In terms of the number of layers in NeRT, we set $L_t$, $L_f$ and $L_s$ to 2, and $L_p$ to 5.

### K.2   ADDITIONAL EXPERIMENTAL RESULTS

Table 16: **Full table on long-term time series.** The best results are reported in boldface.

|  | Drop ratio | Linear | Cubic | SIREN | FFN | NeRT |
|---|---|---|---|---|---|---|
| ETTh1 | 30% | 0.0892 | 0.1268 | $0.1945\pm0.0030$ | $0.2522\pm0.0392$ | $\mathbf{0.0828}\pm\mathbf{0.0028}$ |
|  | 50% | 0.1178 | 0.1662 | $0.2173\pm0.0216$ | $0.3407\pm0.0133$ | $\mathbf{0.0911}\pm\mathbf{0.0097}$ |
|  | 70% | 0.1978 | 0.2902 | $0.2605\pm0.0082$ | $0.4256\pm0.0199$ | $\mathbf{0.1257}\pm\mathbf{0.0056}$ |
| ETTh2 | 30% | 0.0407 | 0.0655 | $0.1010\pm0.0075$ | $0.1863\pm0.0113$ | $\mathbf{0.0344}\pm\mathbf{0.0020}$ |
|  | 50% | 0.0473 | 0.0847 | $0.0723\pm0.0021$ | $0.2351\pm0.0138$ | $\mathbf{0.0423}\pm\mathbf{0.0022}$ |
|  | 70% | 0.0596 | 0.1250 | $0.0964\pm0.0024$ | $0.3178\pm0.0460$ | $\mathbf{0.0575}\pm\mathbf{0.0001}$ |
| National Illness | 30% | 0.0266 | 0.0248 | $0.3502\pm0.0210$ | $0.1110\pm0.0059$ | $\mathbf{0.0239}\pm\mathbf{0.0109}$ |
|  | 50% | 0.0567 | 0.0484 | $0.1716\pm0.0376$ | $0.2319\pm0.0737$ | $\mathbf{0.0291}\pm\mathbf{0.0048}$ |
|  | 70% | 0.0902 | 0.0876 | $0.3564\pm0.0202$ | $0.4453\pm0.0442$ | $\mathbf{0.0871}\pm\mathbf{0.0257}$ |

We report the full experimental results of Table 2 in Table 16. As shown in Table 16, our NeRT shows the lowest MSE in every dataset, regardless of the drop ratio. For example, NeRT shows an MSE of 0.1257 in ETTh1 with a drop ratio of 70%, while baselines exhibit errors from 0.1978 in minimum to 0.4256 in maximum. Figures 23, 24, and 25 show how models learn and represent the spatiotemporal coordinates of ETTh1, ETTh2, and National Illness, respectively. In those figures, the top row ((a)-(c)) distinguishes the training and the testing sets in the learned coordinate systems where the X-axis refers to the temporal information and the Y-axis is the spatial information. While training, models only see the white-colored coordinates, i.e., training samples, and then predict values in the black-colored coordinates, i.e., validating and testing samples. Other rows are the results of the first 50 timestamps by each method in each dataset. Unlike other baselines, NeRT successfully learns the spatial coordinate systems to embed features and accurately represents the temporal pattern in each feature. Surprisingly, NeRT demonstrates remarkable predictions even in extreme scenarios with a drop ratio of 70%, and it maintains its performance well compared to other baselines in challenging situations, i.e., high drop ratios.

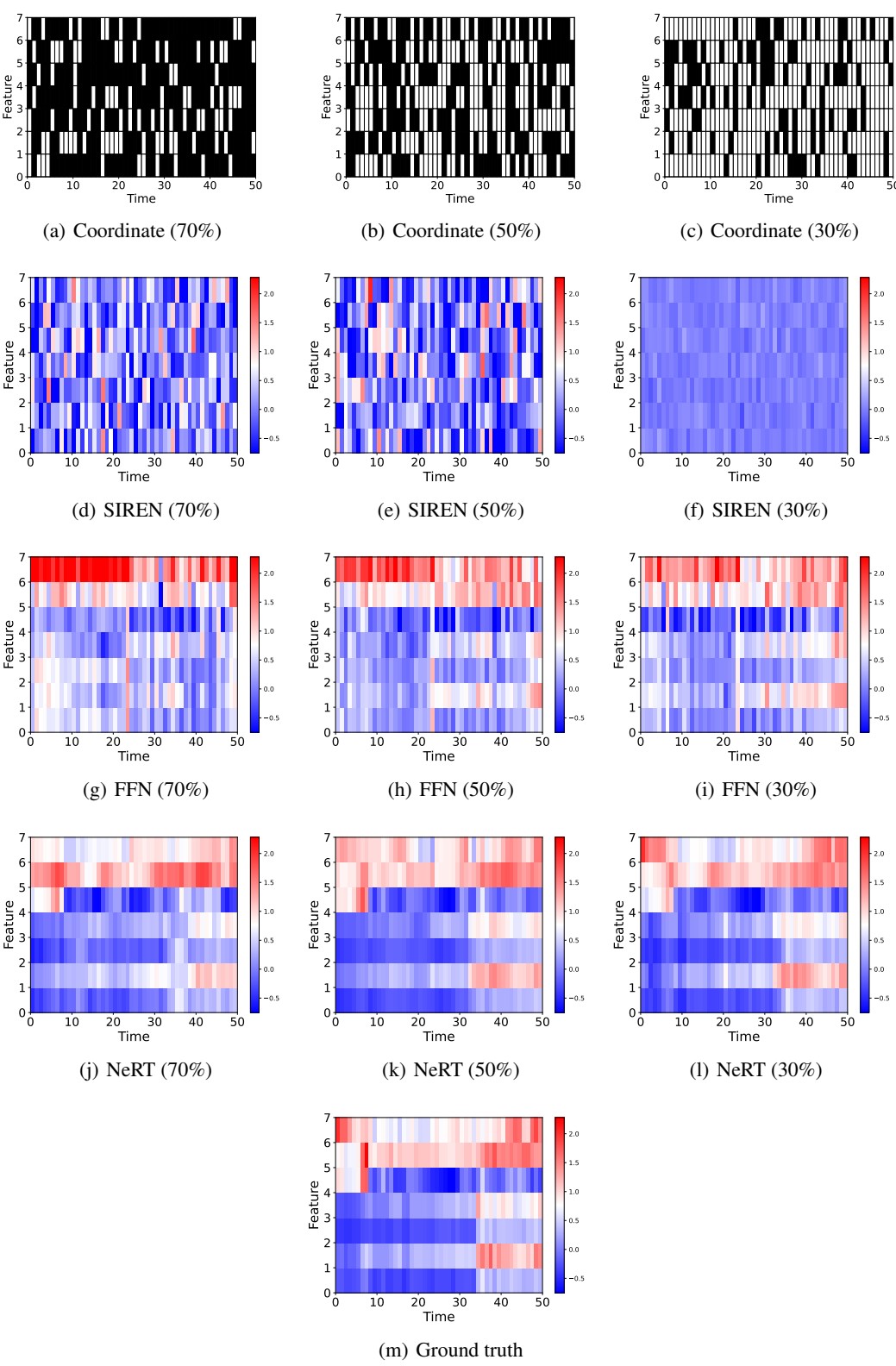

(a) Coordinate (70%)    (b) Coordinate (50%)    (c) Coordinate (30%)

(d) SIREN (70%)    (e) SIREN (50%)    (f) SIREN (30%)

(g) FFN (70%)    (h) FFN (50%)    (i) FFN (30%)

(j) NeRT (70%)    (k) NeRT (50%)    (l) NeRT (30%)

(m) Ground truth

Figure 23: Experimental results of long-term time series (ETTh1). In (a)-(c), white (resp. black) cells mean training (resp. validating/testing) samples.

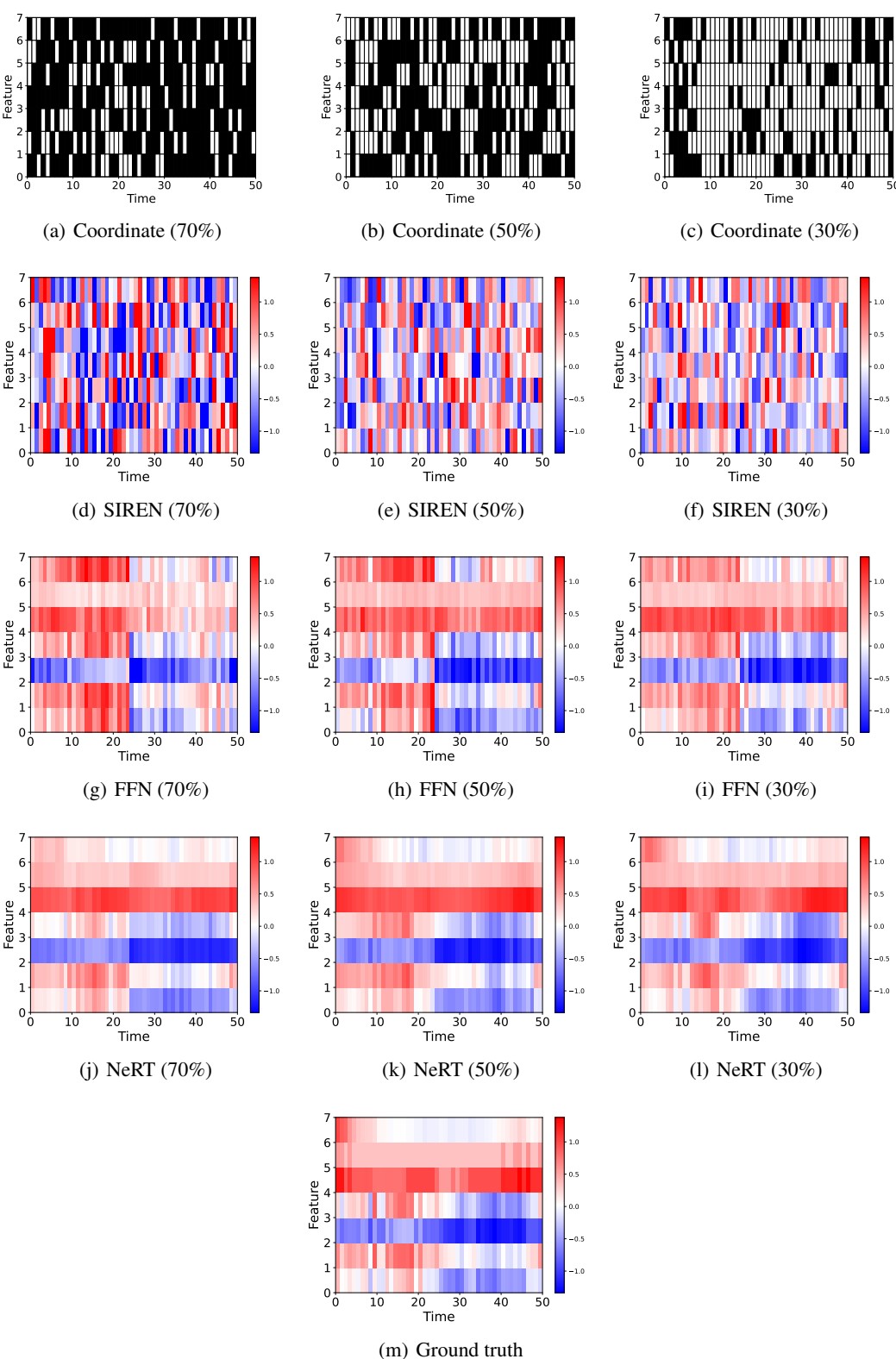

Figure 24: Experimental results of long-term time series (ETTh2). In (a)-(c), white (resp. black) cells mean training (resp. validating/testing) samples.

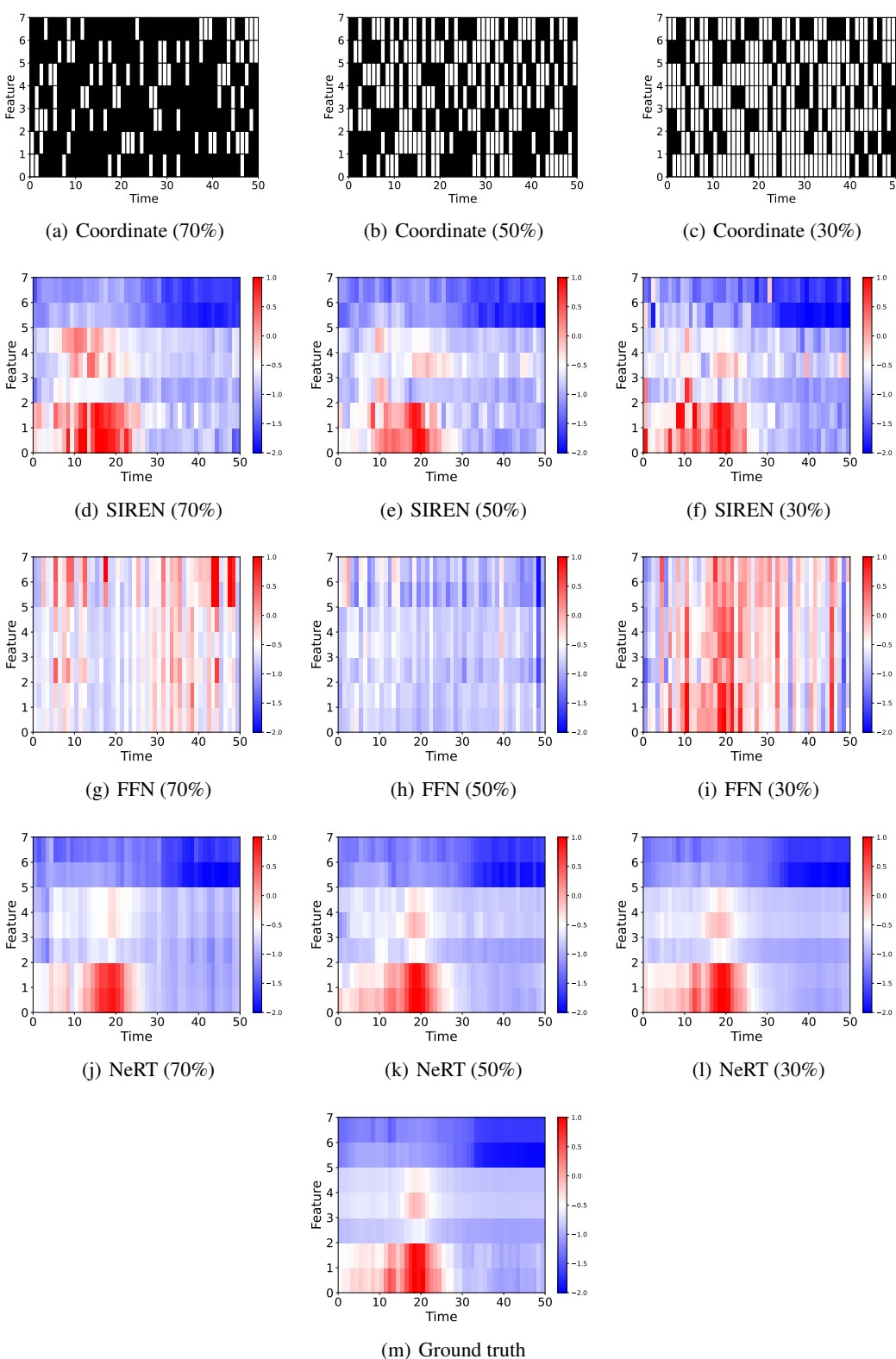

Figure 25: Experimental results of long-term time series (National Illness). In (a)-(c), white (resp. black) cells mean training (resp. validating/testing) samples.

