# OpenReview forum: "NeRT: Implicit Neural Representation for Time Series"
_ICLR.cc/2024/Conference — ICLR 2024 Conference Withdrawn Submission_

### Official Review · Reviewer_9g36 · 2023-10-29

**Soundness:** 4 excellent
**Presentation:** 3 good
**Contribution:** 3 good
**Rating:** 6
**Confidence:** 4

**Summary:**

This paper proposes a new neural implicit representation model for time series data. The model has an encoder-decoder architecture, a novel learnable Fourier mappings and a spatiotemporal representation. The output is a multiplication of scale component and periodic component. Experiments show better performance than SIREN and FFN.

**Strengths:**

1. This paper supports both imputation and extrapolation.
2. Ablation studies show that the proposed components (spatiotemporal representation, learnable Fourier transform) do lead better performance.

**Weaknesses:**

1. In the intro section, the claim that Transformers reply on regularly sampled time-series data is wrong. For example, [1] shows that the Transformer model handles irregularly-sampled time series well for imputation.

2. In section 2, "Finally, INRs operate on a per-data-instance basis, meaning that one time-series instance is required to train an INR". This claim is true but I don't think it is an advantage. A model that can only handle a single time series data is almost useless.

3. In section 3.1, why $c_i^t$ is a vector rather than a scalar? It just denotes a temporal coordinate of a time and should be a scalar.

4. Why the frequency of the Fourier feature is uniformly sampled?

5. "Vectors Bm ∈ R1×DψF , δm ∈ R1×DψF denote the phase shift and the bias respectively". I think what the author wants to claim is that  B represents bias while δm denotes phase shift.

6. In Figure 5, it is unclear what the vertical axis represents.

[1] NRTSI: Non-Recurrent Time Series Imputation, ICASSP 2023.

**Questions:**

None

---

### Official Review · Reviewer_96cx · 2023-10-30

**Soundness:** 1 poor
**Presentation:** 2 fair
**Contribution:** 2 fair
**Rating:** 3
**Confidence:** 4

**Summary:**

The paper proposes to leverage implicit neural representations (INRs) to model time series in order to overcome the challenges of irregular measurements and the dependency on the window size. The model explores the block interpolation and the forecasting tasks. The model relies on one hand on a time coordinate system which maps each timestamp $t_{i}$ to a vector $c_{i}^{t}$ representing the calendar information (second, minute, hour etc.). On the other hand, the model utilizes a one hot encoding to encode the feature index of the considered time series instance. The feature index coordinates are passed through a feature encoder which is a classical composition of MLP's. The time coordinate is passed simultaneously through a classical composition of MLP's and a composition of learnable Fourier feature mapping on the other hand (in order to learn the time series frequencies). Then the embedding vectors are passed through MLP's decoders to produce a scale / factor decomposition. In the experiments, the INR baselines consist of vanillas SIREN and FFN. Experiments are performed on different time series and on a PDE with periodic behaviors, namely the 2D-Helmholtz equation.

**Strengths:**

- NeRT can handle irregular time series and perform joint interpolation and forecasting, which opens interesting applications.

- NeRT architecture is inspired by a classic time series decomposition into scale and periodic components, which allows some comprehensive insights in some cases.

**Weaknesses:**

### Architecture weaknesses.
NeRT relies on the coordinate based system which informs the network about the important frequencies and the considered time series instance. It implies two major flaws :

- This architecture assumes have access to the calendar information, which is not standard in the deep learning time series literature. In addition, the calendar information is not always available in the real-world problem application. % In my opinion a general Deep Learning model for time series can not based its performances on this kind of information.
- Secondly, the training procedure according to the coordinate system is not clear to me. If I am not mistaken, the training of NeRT consists in just fitting the observed points and then query any timestamp $t$ in the defined coordinate system. This property implies that NeRT doesn't rely on a given look-back window but also implies that NeRT doesn't adapt to new context. In practice, we want a forecaster to be reusable in the future and I cannot understand how NeRT can give decent performances by not considering a new context/look-back window when dealing with a new forecasting time window.

### Experiments weaknesses.
It is impossible, in the current state, to have any certainty about the performances of the model, as explained below.

First, in the main experiment content, the only baselines are a vanilla SIREN and a FFN. It is not clear to me how both vanilla INR architectures make use of the coordinate system and how they are used during the training and inference phases. The Figure 4 is also quite surprising, I wonder why FFN underfits that much in cases (b) and (e) and why SIREN performs so poorly in the forecasting task in case (d).

From the main experiment section, it is impossible to assess the quality of NERT. Fortunately, authors compare to classical forecasting methods in Appendix I and linear and cubic interpolation in H.2. However these comparisons seem to be unfair because of the followings:

- First, for the forecasting experiments in Appendix I, authors compare to several baselines which don't make use at all of the calendar information. This is problematic since giving the time coordinate system results in giving explicitly the right frequencies and seasonality for datasets like \textit{Electricity}, \textit{Traffic}, \textit{Caiso} and \textit{NP}.
- Moreover, classical baselines such as DLinear are trained with a look-back window of a length of at most 192. It is more standard to take longer look-back windows as demonstrated in [1, 2]. For instance, in Figure 4 of [1], the authors demonstrate that for horizon 720, the results increase significantly with a look-window of length 672 or 720 compared to small look-back windows. This results in underfiting the baselines as shown in Figure 22.
In addition, another major issue emerges from the training of these methods. All theses methods are trained according to a drawing of several pairs of (look-back window, horizon) in the training window. This means that for instance for horizon 96 and look-back window of length 192, the draw window is 288-long. On the other hand, if I understand correctly the training procedure described in Appendix H.1., NeRT is fitted on each yellow block which consist in a total of 2500 timestamps. This difference in the training procedure does not seem to be fair.
- Lastly, in the block imputation experiments, in Table 9, it is hard to assess the quality of NeRT because the linear interpolation and the cubic interpolation are two poor baselines. The best of both baselines is the linear interpolation which is a dummy constant for 500 timestamps.

### Narrative flaws and weaknesses.
- This work is not the first one to use to INRs to model time series. DeepTime [3], which learns a set of basis INR functions and combines them using a Ridge regressor to mainly perform time series forecasting, is cited but there is no real discussion on this important concurrent work. Moreover, since this is a INR continuous-time baseline, it should be included with the naive SIREN and FFN implementations.
- There exists other continuous-time Deep Learning methods that can handle irregular measurements and perform both imputation and forecasting, such as neural processes [4]. And DeepTime, although this was not tested in the original paper, can also perform both tasks. In addition, there are works on unified models for time series imputation and forecasting (for \eg [5, 6]). So the statement L3 , \ie 'non-existence of unified model for time series forecasting and Imputation' is not true, since there are discrete and continuous models being able to perform both.
- The statement 'Transformer-based time series models are sometimes surprisingly worse than simple Linear models' is not true anymore, since PatchTST [2] showed that transformers were not applied efficiently, using attention between timestamps, and that using attention between patches of sequences was more intuitive and efficient. Thus, the statement 'large models are quickly overfitted and their testing accuracy become mediocre' does not stand. It would be then very important to have the state-of-art method forecasting method, PatchTST [2], as a baseline.

[1]: Zeng, A., Chen, M., Zhang, L., Xu, Q. (2023, June). Are transformers effective for time series forecasting?. In Proceedings of the AAAI conference on artificial intelligence (Vol. 37, No. 9, pp. 11121-11128).

[2]: Nie, Y., Nguyen, N. H., Sinthong, P., Kalagnanam, J. (2022). A time series is worth 64 words: Long-term forecasting with transformers. arXiv preprint arXiv:2211.14730.

[3]: Gerald Woo, Chenghao Liu, Doyen Sahoo, Akshat Kumar, and Steven Hoi. Deeptime: Deep time-
index meta-learning for non-stationary time-series forecasting. arXiv preprint arXiv:2207.06046,
2022.

[4]: M. Garnelo, D. Rosenbaum, C. Maddison, T. Ramalho, D. Saxton, M. Shanahan, Y. W. Teh, D. J.
Rezende, and S. M. A. Eslami. Conditional neural processes. In Proceedings of the 35th Interna-
tional Conference on Machine Learning, ICML, volume 80, pages 1690–1699. PMLR, 2018.

[5]: Park S, Yoon S, Lee B, Ko S, Hwang E. Probabilistic forecasting based joint detection and imputation of clustered bad data in residential electricity loads. Energies. 2020.

[6]: Tran TH, Nguyen LM, Yeo K, Nguyen N, Phan D, Vaculin R, Kalagnanam J. An End-to-End Time Series Model for Simultaneous Imputation and Forecast. arXiv preprint arXiv:2306.00778. 2023.

**Questions:**

- "We design the experiment using the first 10 samples" P.21: Why use only 10 samples for the periodic time series datasets? Electricity and Traffic have a few hundred of samples each.
- In Figures 9-11, could you explain why the scale factor increases over time whereas NeRT seems to produce constant amplitude values which corresponds to the solution? In contrast, in Figures 11-12, it seems coherent since there is damping and the scale factor goes down with the time.

---

### Official Review · Reviewer_wYL3 · 2023-11-04

**Soundness:** 2 fair
**Presentation:** 3 good
**Contribution:** 2 fair
**Rating:** 5
**Confidence:** 4

**Summary:**

This manuscript introduces implicit neural representations (INRs) for time series. The proposed model includes a new coordinate system with a learnable Fourier feature mapping and generates periodic and scale components of time series. Experiments are mainly conducted on time series interpolation and extrapolation tasks.

**Strengths:**

It is interesting to study spatiotemporal coordinates for multivariate time series INR.

**Weaknesses:**

1. The spatiotemporal coordinates are developed for time series INR. However, the experiments only focus on univariate time series. Additionally, it is unclear why the standard INR cannot be used for multivariate time series (in my opinion, MTS just needs time coordination and multiple outputs in INR). It should be noted that the original INR has been widely utilized for parameterizing spatially dependent or time-dependent PDEs. Could you clarify these issues?
2. While INR is primarily trained on a single time series sample and can impute missing values or predict subsequent values, it is unclear how to apply the proposed method in the case of multiple samples. Could you elaborate on this issue and explain how the proposed method can be extended to handle multiple time series samples?
3. The investigation and discussion on related works of using INR for time series data are insufficient. For example, the following recent papers also explore the use of INR for time series analysis: '2022 Time-Series Anomaly Detection with Implicit Neural Representation,' '2023 DeepTIMe Deep Time-Index Meta-Learning for Non-Stationary Time-Series Forecasting,' and '2023 Time Series Continuous Modeling for Imputation and Forecasting with Implicit Neural Representations.'
4. The experimental setup for comparing the proposed method and windowing-based baselines is unclear. Please provide more details on the experimental design.
5. Please discuss the potential limitations of using INR for time series and whether it is necessary to train an INR for each time series sample.
6. In Figure 22, it is surprising that strong baselines such as DLinear and Informer cannot learn basic periodic patterns in traffic datasets. Could you explain why this is the case and how to determine the appropriate window sizes for various baselines?

**Questions:**

Please help address my questions in the Weaknesses part.

---

### Official Review · Reviewer_34Ud · 2023-11-09

**Soundness:** 2 fair
**Presentation:** 2 fair
**Contribution:** 1 poor
**Rating:** 3
**Confidence:** 4

**Summary:**

This paper introduces a novel approach using Implicit Neural Representations (INRs) to model time series data. This method addresses the challenges of irregular measurements and the need to define specific time windows. The model is mainly evaluated on block interpolation and forecasting tasks.

To achieve this, the model employs two different coordinate projections :
* A time coordinate system that maps each timestamp to a vector that encodes calendar information.
* A one-hot encoding to represent the feature index of each sample.

Then, each coordinate projection is passed separably through MLPs. One of the MLPs that takes the time coordinate embedding as input can learn frequency through the Fourier mapping layer (basically a learnable SIREN layer). After the encoding phase, the encoder outputs are recombined in the latent space before being passed to MLP decoders. The outputs of the decoders are recombined in a scale/factor decomposition to allow some interpretability.

The NeRT Architecture is mainly evaluated against vanilla SIREN and Fourier Features network.

**Strengths:**

* Adapting INRs for time series is an interesting approach, offering a way to model continuously time series data.
* The simultaneous execution of block imputation and forecasting adds an interesting dimension to the paper.
* The scale/factor decomposition aspect might provide valuable insights.

**Weaknesses:**

**Major Weaknesses:**

* The paper is difficult to follow. There is no clear position according to existing continuous methods, and the presentation of the model and its optimization is unclear.
* It's impossible to judge the quality of the model just by comparing it to SIREN and Fourier Features networks. Also, it's unclear to me how SIREN and Fourier Features Network take advantage of the calendar information you provide.
* In the appendix, you compare NeRT to other forecasting models that do not use calendar information. The comparison doesn't seem fair.
* The fact that you fit the entire period doesn't seem appropriate for the distribution shift that often occurs in time series forecasting. Thus, not relying on a specific time window can be a strength, but it can also hurt the model performance for future forecasts.


**Minor Weaknesses:**

* Some important baselines are missing (such as Patch-TST), and Dlinear appears to be underfitted (as shown in Figure 22) because of the short look-back windows you choose.
* Cubic interpolation and linear interpolation are not serious baselines for block interpolation.
* I do not understand the 2D Helmholtz experiment in a time series paper.
* DeepTime, which is a strong INR baseline, can also do forecasting and imputation.

**Questions:**

* Q.1. Can you explain in more detail way the training procedure?
* Q.2. How does the FFN et SIREN baseline take into account the calendar information?
* Q.3. How does NeRT perform against other forecasting baselines if you do not provide calendar information (and rely only on the learnable Fourier mapping to adapt to the right frequencies)?
* Q.4. In the block imputation task, how does NeRT perform compared to a simple « repeat the last block » baseline?